# Trisomy 21 consistently activates the interferon response

**Kelly D Sullivan**[1,2,3,4]*, **Hannah C Lewis**[1,2], **Amanda A Hill**[1,2], **Ahwan Pandey**[1,2,3,4], **Leisa P Jackson**[1,3,4], **Joseph M Cabral**[1,3,4], **Keith P Smith**[1], **L Alexander Liggett**[1,5], **Eliana B Gomez**[1,3,4], **Matthew D Galbraith**[1,2,3,4], **James DeGregori**[1,5,6,7,8,9], **Joaquín M Espinosa**[1,2,3,4]*

[1]Linda Crnic Institute for Down Syndrome, University of Colorado School of Medicine, Aurora, United States; [2]Department of Pharmacology, University of Colorado School of Medicine, Aurora, United States; [3]Department of Molecular, Cellular and Developmental Biology, University of Colorado Boulder, Boulder, United States; [4]Howard Hughes Medical Institute, Chevy Chase, United States; [5]Department of Biochemistry and Molecular Genetics, University of Colorado School of Medicine, Aurora, United States; [6]Department of Pediatrics, University of Colorado School of Medicine, Aurora, United States; [7]Integrated Department of Immunology, University of Colorado School of Medicine, Aurora, United States; [8]Section of Hematology, University of Colorado School of Medicine, Aurora, United States; [9]Department of Medicine, University of Colorado School of Medicine, Aurora, United States

*For correspondence: kelly.d. sullivan@ucdenver.edu (KDS); joaquin.espinosa@ucdenver.edu (JME)

**Abstract** Although it is clear that trisomy 21 causes Down syndrome, the molecular events acting downstream of the trisomy remain ill defined. Using complementary genomics analyses, we identified the interferon pathway as the major signaling cascade consistently activated by trisomy 21 in human cells. Transcriptome analysis revealed that trisomy 21 activates the interferon transcriptional response in fibroblast and lymphoblastoid cell lines, as well as circulating monocytes and T cells. Trisomy 21 cells show increased induction of interferon-stimulated genes and decreased expression of ribosomal proteins and translation factors. An shRNA screen determined that the interferon-activated kinases JAK1 and TYK2 suppress proliferation of trisomy 21 fibroblasts, and this defect is rescued by pharmacological JAK inhibition. Therefore, we propose that interferon activation, likely via increased gene dosage of the four interferon receptors encoded on chromosome 21, contributes to many of the clinical impacts of trisomy 21, and that interferon antagonists could have therapeutic benefits.

## Introduction

Trisomy 21 (T21) is the most common chromosomal abnormality in the human population, occurring in approximately 1 in 700 live births (*Alexander et al., 2016*). The extra copy of chromosome 21 (chr21) impacts human development in diverse ways across every major organ system, causing the condition known as Down syndrome (DS). One of the most intriguing aspects of T21 is that it causes an altered disease spectrum in the population with DS, protecting these individuals from some diseases (e.g. solid tumors, hypertension), while strongly predisposing them to others (e.g. Alzheimer's disease, leukemia, autoimmune disorders) (*Alexander et al., 2016*; *Sobey et al., 2015*; *Bratman et al., 2014*; *Roberts and Izraeli, 2014*; *Anwar et al., 1998*; *Malinge et al., 2013*; *Hasle et al., 2016*). Despite many years of study, the molecular, cellular, and physiological

**eLife digest** Our genetic information is contained within structures called chromosomes. Down syndrome is caused by the genetic condition known as trisomy 21, in which a person is born with an extra copy of chromosome 21. This extra chromosome affects human development in many ways, including causing neurological problems and stunted growth. Trisomy 21 makes individuals more susceptible to certain diseases, such as Alzheimer's disease and autoimmune disorders – where the immune system attacks healthy cells in the body – while protecting them from tumors and some other conditions.

Since cells with trisomy 21 have an extra copy of every single gene on chromosome 21, it is expected that these genes should be more highly expressed – that is, the products of these genes should be present at higher levels inside cells. However, it was not clear which genes on other chromosomes are also affected by trisomy 21. Sullivan et al. aimed to identify which genes are affected by trisomy 21 by studying samples collected from a variety of individuals with, and without, this condition.

Four genes in chromosome 21 encode proteins that recognize signal molecules called interferons, which are produced by cells in response to viral or bacterial infection. Interferons act on neighboring cells to regulate genes that prevent the spread of the infection, shut down the production of proteins and activate the immune system. Sullivan et al. show that cells with trisomy 21 produce high levels of genes that are activated by interferons and lower levels of genes required for protein production. In other words, the cells of people with Down syndrome are constantly fighting a viral infection that does not exist.

Constant activation of interferon signaling could explain many aspects of Down syndrome, including neurological problems and protection against tumors. The next steps are to fully define the role of interferon signaling in the development of Down syndrome, and to find out whether drugs that block the action of interferons could have therapeutic benefits.

mechanisms driving both the protective and deleterious effects of T21 are poorly understood. A few chr21-encoded genes have been implicated in the development of specific comorbidities, such as *APP* in Alzheimer's disease (*Wiseman et al., 2015*), and *DYRK1A* and *ERG* in hematopoietic malignancies (*Stankiewicz and Crispino, 2013*; *Malinge et al., 2012*). Therefore, research in this area could inform a wide range of medical conditions affecting not only those with DS, but also the typical population.

The clinical manifestation of DS is highly variable among affected individuals, with various comorbidities appearing in a seemingly random fashion, suggesting the presence of strong modifiers, genetic or otherwise, of the deleterious effects of T21. Even conserved features, such as cognitive impairment, display wide quantitative variation (*de Sola et al., 2015*). Collectively, our understanding of the mechanisms driving such inter-individual variation in the population with DS is minimal. More specifically, it is unclear what gene expression changes are consistently caused by T21, versus those that are context-dependent. Integrated analyses of a large body of studies have indicated that the changes in gene expression caused by T21 involve various signaling pathways (*Scarpato et al., 2014*), however, these studies vary widely in cell type, number of samples, and even analysis platform, among other variables (*Volk et al., 2013*; *Costa et al., 2011*). More recently, gene expression analysis of cells derived from discordant monozygotic twins, only one of which was affected by T21, concluded that global gene expression changes in T21 cells are driven by differences in chromatin topology, whereby affected genes are clustered into large chromosomal domains of activation or repression (*Letourneau et al., 2014*). However, independent re-analysis of these data has challenged this conclusion (*Do et al., 2015*). Therefore, there remains a clear need to identify the consistent gene expression changes caused by T21 and to characterize how these programs are modified across cell types, tissue types, genetic backgrounds, and developmental stages.

In order to identify *consistent* signaling pathways modulated by T21, defined as those that withstand the effects of inter-individual variation, we employed two complementary genomics approaches, transcriptome analysis and shRNA loss-of-function screening, in both panels of cell lines

and primary cell types from individuals of diverse genetic background, gender, and age, with and without T21. Our RNA-seq transcriptome analysis identified *consistent* gene expression signatures associated with T21 in all cell types examined. Interestingly, the fraction of this gene expression signature that is not encoded on chr21 is dominated by the interferon (IFN) transcriptional response, an observation that is reproducible in skin fibroblasts, B cell-derived lymphoblastoid cell lines, as well as primary monocytes and T cells. In parallel, we performed a kinome-focused shRNA screen that identified the IFN-activated kinases JAK1 and TYK2 as strong negative regulators of T21 cell proliferation in fibroblasts. Importantly, pharmacological inhibition of JAK kinases improves T21 cell viability. Taken together, our results identify the IFN pathway as *consistently* activated by T21, which could merely be a result of increased gene dosage of four IFN receptor subunits encoded on chr21. We hypothesize that IFN activation could contribute to many of the effects of T21, including increased risk of leukemia and autoimmune disorders, as well as many developmental abnormalities also observed in interferonopathies (*Yao et al., 2010*; *Zitvogel et al., 2015*; *Crow and Manel, 2015*; *McGlasson et al., 2015*).

## Results

### Trisomy 21 causes consistent genome-wide changes in gene expression

In order to investigate *consistent* gene expression signatures associated with T21, we performed RNA-seq on a panel of 12 age- and gender-matched human fibroblasts from euploid (disomic, D21) and T21 individuals (*Figure 1—figure supplement 1A–C*). T21 was confirmed by PCR analysis of the chr21-encoded *RCAN1* gene (*Figure 1—figure supplement 1D*). We included samples from different genetic backgrounds, ages, and genders, in order to avoid identifying differences that are specific to a particular pair of isogenic or genetically related cell lines and which would not withstand the effects of inter-individual variation. To illustrate this point, comparison of one pair of disomic male individuals of similar age yielded thousands of differentially expressed genes (DEGs), with similar numbers of upregulated and downregulated DEGs (*Figure 1A–B*, Male 1 vs. Male 2). However, when the 12 samples are divided into two groups with roughly balanced age, sex, and T21 status, very few consistent changes were identified, thus demonstrating the impact of inter-individual variation within our sample set (*Figure 1A–B*, *Figure 1—figure supplement 1C*, Group 1 vs. Group 2). In contrast, comparison of all T21 versus all D21 cells identified 662 consistent DEGs, with a disproportionate number of these upregulated in T21 cells (471 of 662, *Figure 1A*, T21 vs. D21, *Supplementary file 1A*). We also observed an uncharacteristic spike of DEGs at ~1.5-fold overexpression in T21 cells on a volcano plot, consistent with many chr21 genes being overexpressed solely due to increased gene dosage (*Figure 1B*). For comparison purposes, we also analyzed samples by sex which expectedly yielded DEGs encoded on chrX (e.g. *XIST*) and chrY (*Figure 1 A–B*; Female vs. Male). Sex causes fewer significant changes than T21, with roughly equal numbers of upregulated and downregulated genes. Taken together, these data indicate that T21 produces consistent changes in a gene expression signature that withstands differences in genetic background, age, sex, and site of biopsy. Of note, when we performed RNA-seq analysis using increasing numbers of T21 vs. D21 pairs, the fraction of chr21-encoded DEGs increased steadily with sample size, accounting for ~12% of the core gene expression signature in the 12 cell line panel. However, 88% of DEGs are located on other chromosomes, indicating the existence of conserved mechanisms driving these genome-wide changes in gene expression (*Figure 1—figure supplement 1E*).

A recent report concluded that changes in gene expression caused by T21 between a single pair of discordant monozygotic twins were due to dysregulation of chromosomal domains (*Letourneau et al., 2014*). Thus, we next asked where the ~88% of core DEGs not encoded on chr21 are located across the genome. This exercise revealed broad distribution across all chromosomes, with no obvious contiguous domains of up- or downregulation (see *Figure 1—figure supplement 2A* for a whole genome Manhattan plot, and *Figure 1—figure supplement 3* for individual chromosomes). In fact, mere visual analysis of DEGs from the individual chromosomes previously claimed by Letourneau et al. to harbor large dysregulated domains (e.g. chr3, chr11, chr19) did not reveal such domains in our dataset, showing instead obvious regions of overlapping activation and repression (shaded gray boxes in *Figure 1C*). Thus, our analysis is more consistent with the report that re-analyzed the data in Letournaeu et al. and questioned the existence of these chromosomal

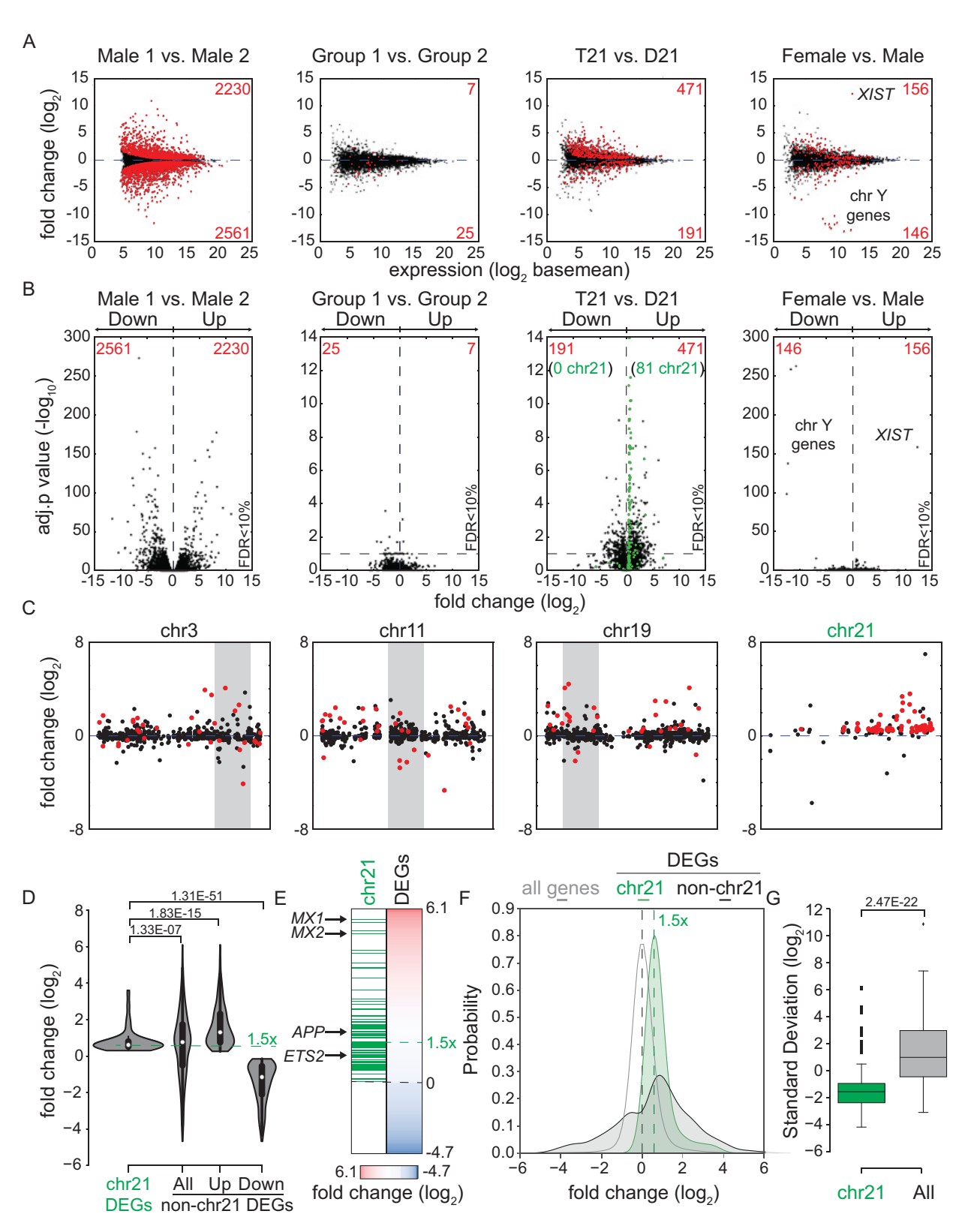

**Figure 1.** Transcriptome analysis identifies consistent changes in global gene expression between euploid (D21) and trisomy 21 (T21) fibroblasts. (A) MA plots displaying the results of RNA-seq analysis for the indicated comparisons (see *Figure 1—figure supplement 1A–C*). Differentially expressed

*Figure 1 continued on next page*

*Figure 1 continued*

genes (DEGs), as defined by DEseq2 (FDR < 10%), are labeled in red. (B) Volcano plots of comparisons in A highlight changes in chr21 gene expression (green) consistent with increased gene dosage effects. (C) Manhattan plots displaying DEGs (red) and all genes (black) for individual chromosomes do not show obvious domains of contiguous upregulation or downregulation. Shaded areas highlight regions of overlapping upregulation and downregulation (see *Figure 1—figure supplements 2A* and *3*). (D) Violin plots of chr21 and non-chr21 DEGs displaying the distribution of fold changes of DEGs in each category. p-values were calculated with the Kolmogorov-Smirnov test. (E) Heatmap of all significant DEGs showing clustering of chr21 DEGs (green) around 1.5 fold upregulation in T21 cells. (F) Kernel density estimate plot highlighting the probabilities of chr21 DEGs (green, green dashed line indicates median), non-chr21 DEGs (black, black dashed line indicates median), and all genes (gray), of having a given fold change. (G) Box and whisker plot of standard deviations of fold changes in DEGs for six pairwise comparisons of age- and gender-matched T21 versus D21 fibroblasts showing greater variation in fold change for non-chr21 DEGs. p-values were calculated with the Kolmogorov-Smirnov test.

The following figure supplements are available for figure 1:

**Figure supplement 1.** T21 and D21 fibroblast RNA-seq.

**Figure supplement 2.** Amplification of changes in gene expression emanating from T21.

**Figure supplement 3.** Differentially expressed genes in trisomy 21 fibroblasts are not organized into obvious chromatin domains.

domains (*Do et al., 2015*). In fact, the only region of the genome at which there was clear contiguous upregulation of DEGs was chr21 itself (*Figure 1C*, *Figure 1—figure supplements 2A* and *3*).

In order to characterize the mechanism driving the consistent changes caused by T21, we examined the regulatory differences between DEGs encoded on chr21 and those not encoded on chr21. Several lines of evidence indicate that, while chr21 DEGs are regulated mostly by increased gene dosage, non-chr21 DEGs may be driven by specific pathways that are subject to signal amplification, with a bias toward upregulation, and greatly affected by inter-individual variation. First, violin plots display the relatively small number of chr21 DEGs, showing mostly upregulation clustered around 1.5 fold, versus a much larger number of non-chr21 DEGs, showing both up- and downregulation with no obvious clustering of fold changes (*Figure 1D*, *Figure 1—figure supplement 2B*). Second, the obvious effect of gene dosage on the expression of chr21 DEGs is apparent in the violin plots and heatmaps (*Figure 1D,E*), where the median fold change centers around 1.5 fold (e.g. *APP*, *ETS2*), while a few genes show greater induction (e.g. *MX1*, *MX2*). In fact, chr21 genes exhibit more than an 80% probability of a ~1.5-fold change as calculated by kernel density estimation analysis (*Figure 1F*). Third, the bias toward upregulation among non-chr21 DEGs is evident in the violin plots, heatmaps, and density estimation analysis (*Figure 1D–F*), where a larger fraction of these genes is upregulated. Finally, we measured the inter-individual variation of chr21 DEGs versus non-chr21 DEGs by calculating the standard deviation for each DEG across each age- and gender-matched pair of fibroblasts. As shown in *Figure 1G*, the median standard deviation for chr21 DEGs is much smaller than for all DEGs.

Altogether, these results suggest the existence of consistent signaling pathways activated by increased dosage of chr21 genes, which in turn cause global changes in gene expression, with a bias toward upregulation and displaying strong inter-individual variation.

## Trisomy 21 leads to constitutive activation of the interferon transcriptional response

Next, we subjected T21 DEGs to upstream regulator analysis using Ingenuity Pathway Analysis (IPA) to identify putative factors contributing to consistent changes in gene expression. This analysis tool includes both a hypergeometric test for overlapping sets of genes and a directional component to predict activation or inactivation of factors that control gene expression (e.g. transcription factors, protein kinases) (*Krämer et al., 2014*). We confirmed the effectiveness of this tool using published RNA expression datasets from our lab for cells treated with an inhibitor of the p53-MDM2 interaction, hypoxia, and serum stimulation (*Sullivan et al., 2012*; *Donner et al., 2010*; *Galbraith et al., 2013*). IPA effectively identified p53, the Hypoxia Inducible Factor 1A (HIF1A), and growth factor receptors and downstream kinases (PDGF, ERK) as the top upstream regulators in each scenario, respectively (*Figure 2—figure supplement 1A*). Strikingly, the top 13 upstream regulators predicted

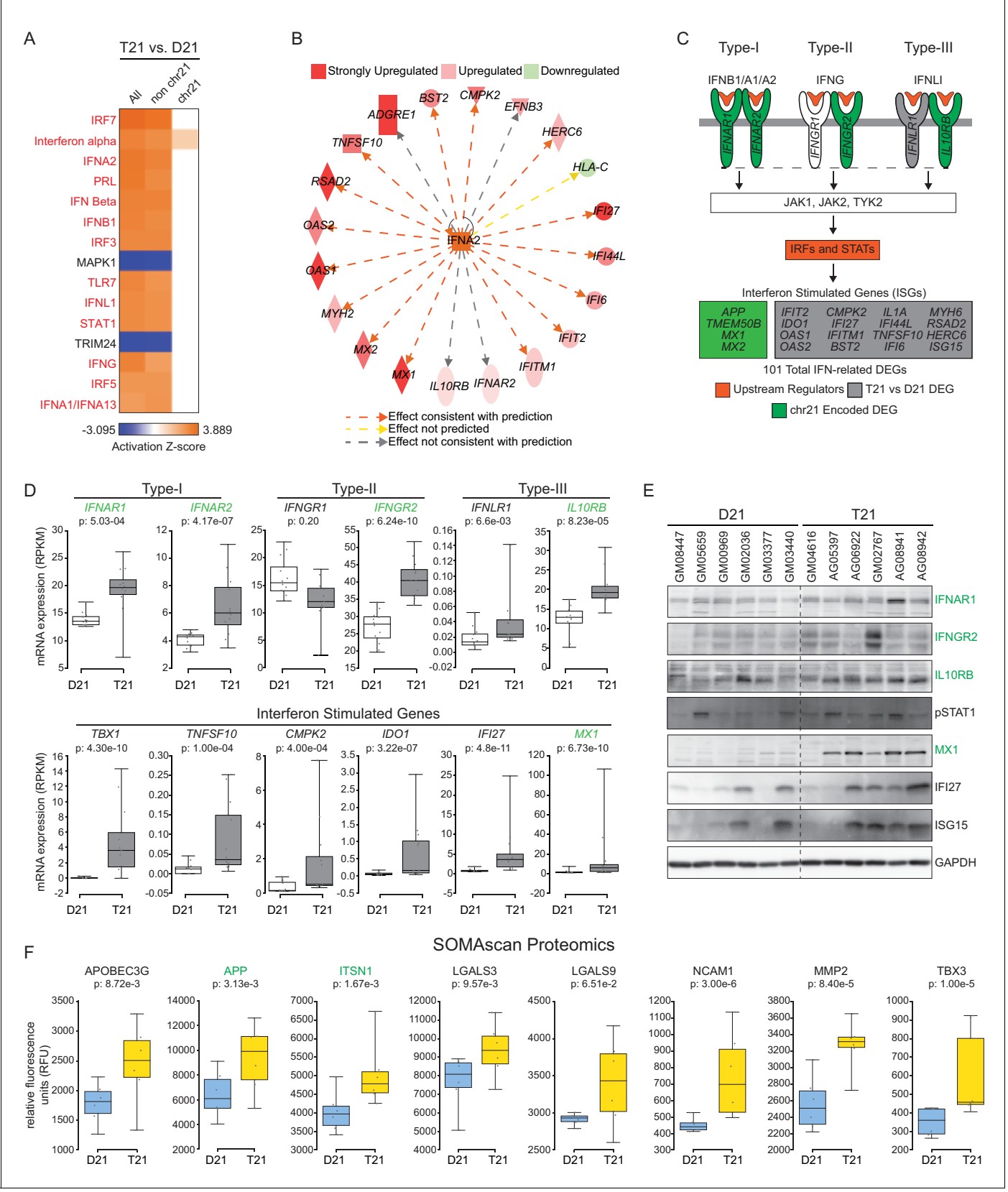

**Figure 2.** The interferon (IFN) transcriptional response is activated in trisomy 21 (T21) fibroblasts. (**A**) Upstream regulator analysis of the T21-associated gene expression signature using Ingenuity Pathway Analysis (IPA) predicts numerous IFN-related factors as activated in T21 cells. (**B**) Representative

*Figure 2 continued on next page*

*Figure 2 continued*

results of the upstream regulator analysis for the Type I IFN ligand IFNA2. (**C**) Graphical summary of the observed deregulation of the IFN pathway in T21 fibroblasts, showing the six IFN receptor subunits, four of which are encoded on chr21 and significantly upregulated in T21 fibroblasts; the predicted upstream regulators (orange), including the Type I, II, and III IFN ligands, as well as the IFN-activated transcription factors (IRFs and STATs); and select examples of Interferon Stimulated Genes (ISGs) upregulated in T21 fibroblasts, either encoded on chr21 (green) or elsewhere in the genome (gray). (**D**) Box and whisker plots showing RNA expression for the six IFN receptor subunits and select ISGs. chr21-encoded genes are highlighted in green. mRNA expression values are displayed in reads per kilobase per million (RPKM). Benjamini-Hochberg adjusted p-values were calculated using DESeq2. (**E**) Western blot analysis confirming upregulation of IFN receptors, STAT1 phosphorylation, and ISGs, in T21 fibroblasts. (**F**) Box and whisker plots showing protein expression of select IFN-related genes as measured by SOMAscan assay. chr21-encoded genes are highlighted in green. Protein expression values are displayed in relative fluorescence units (RFU). Adjusted p-values were calculated using the Empirical Bayes method in QPROT.

The following figure supplement is available for figure 2:

**Figure supplement 1.** Network analysis confirms IFN activation signature in T21 cells.

to be activated in T21 cells are all IFN-related factors, including IFN ligands (e.g. IFNA2, IFNB, IFNG) and IFN-activated transcription factors (e.g. IRF3, IRF5, IRF7, STAT1) (*Figure 2A*). Importantly, most of these signals are derived from non-chr21 DEGs, and would be missed by analyses focused specifically on chr21-encoded genes (*Figure 2A*). This analysis also identified two known repressors of IFN signaling, MAPK1 and TRIM24, as upstream regulators inactivated in T21 cells, consistent with activation of the IFN pathway (*Huang et al., 2008*; *Tisserand et al., 2011*). As an example of how the RNA-seq data supports the upstream regulator prediction by IPA, *Figure 2B* shows the gene network centered on the ligand IFNA2 as a potential driver of consistent gene expression changes. Strong activation of the IFN pathway was also predicted using a different tool, the Pathway Commons Analysis in WebGestalt (*Zhang et al., 2005*; *Wang et al., 2013*; *Cerami et al., 2011*), where 4 of the top 15 pathways identified were IFN-related (*Figure 2—figure supplement 1B*).

Notably, activation of IFN signaling in T21 cells could be explained by the fact that four of the six IFN receptors, *IFNAR1, IFNAR2, IFNGR2*, and *IL10RB*, (representing each IFN class, Type-I, -II, and -III), are chr21-encoded DEGs (*Figure 2C,D*). Using a combination of IPA upstream regulator predictions and our RNA-seq data, we clearly identified the canonical IFN pathways –from ligands through receptors and kinases and down to transcription factors and IFN-stimulated genes (ISGs)– as activated in T21 cells (*Figure 2C*). Whereas IFN receptors are upregulated ~1.5 fold with relatively low levels of inter-individual variation, as expected for increased gene dosage in T21 cells, the downstream ISGs exhibit larger fold changes, greater variation between samples, and tend to have low expression levels in D21 cells, in accord with activation of IFN only in T21 cells (*Figure 2D*). We confirmed the elevated basal expression of three of the IFN receptors (IFNAR1, IFNGR2, and ILR10RB), enhanced basal phosphorylation of STAT1, as well as increased basal expression of several ISGs at the protein level in T21 cells, with noticeable inter-individual variation (*Shuai et al., 1994*; *Waddell et al., 2010*; *Schoggins et al., 2011*) (*Figure 2E*).

We next analyzed protein lysates from the 12 fibroblast lines using SOMAScan technology, which employs DNA aptamers to monitor epitope abundance (*Gold et al., 2012*; *Mehan et al., 2014*; *Hathout et al., 2015*). This assay confirmed elevated protein levels for many of the IFN-related genes found to be induced at the mRNA level in the RNA-seq experiment (*Figure 2F*). Finally, we examined the fraction of our upregulated DEGs linked to IFN signaling using IPA, Pathway Commons, and a list of 387 validated ISGs curated by Schoggins and colleagues (*Schoggins et al., 2011*). Our analysis revealed that 21% (101/471) of DEGs upregulated in T21 cells are linked to IFN signaling, with contributions from both chr21 (17%, 14/81) and non-chr21 (22%, 87/390) DEGs, pointing to IFN activation as a potential mechanism for the larger number of upregulated versus downregulated DEGs (*Figure 2—figure supplement 1C*). Altogether, these results indicate that the IFN pathway is consistently induced by trisomy 21 in fibroblasts, and that the IFN transcriptional response accounts for a considerable fraction of the transcriptome changes caused by trisomy 21 across the genome.

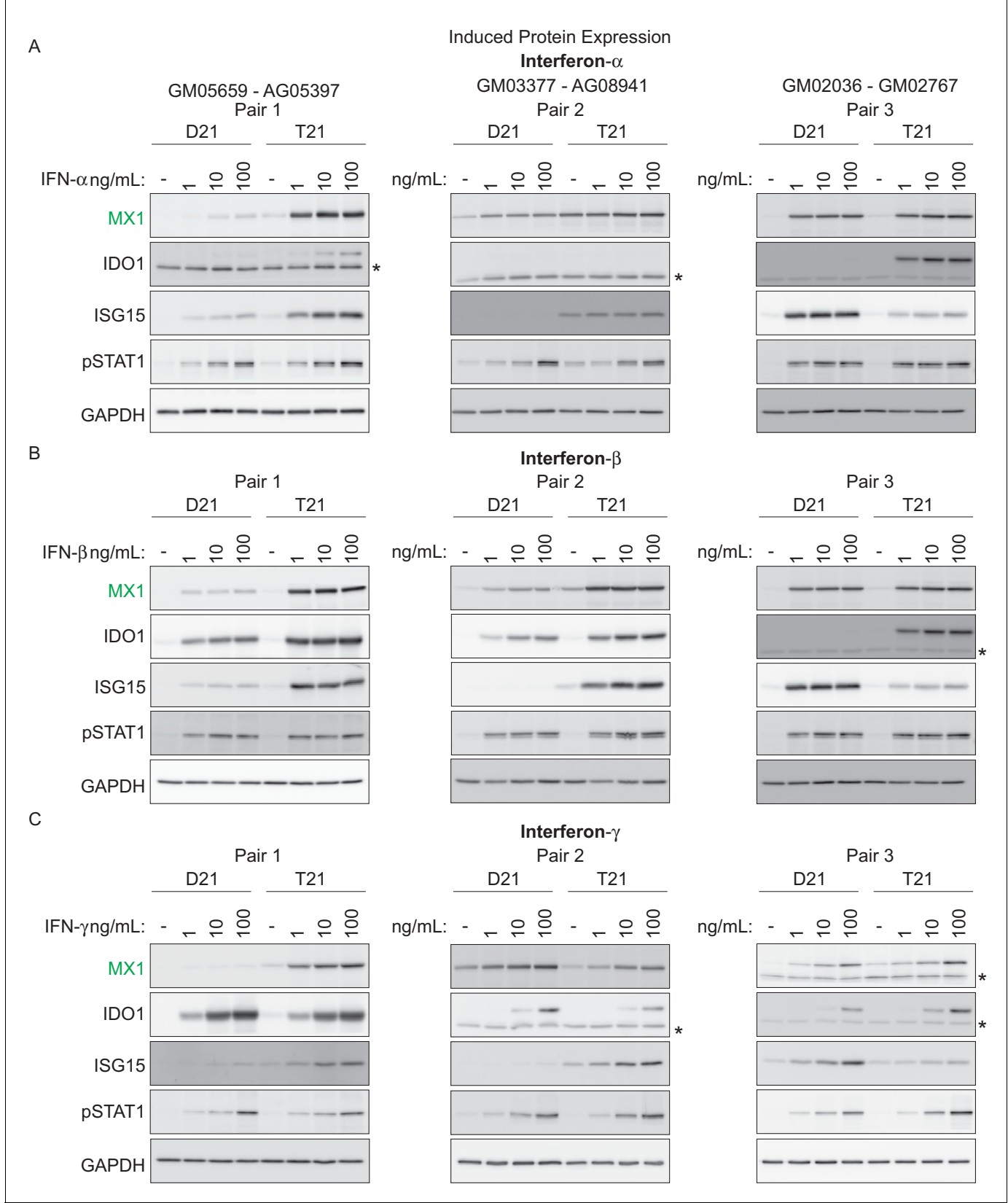

**Figure 3.** T21 fibroblasts are more sensitive to IFN stimulation than D21 fibroblasts. (**A**) Western blots showing that three T21 cell lines are more sensitive to IFN-α treatment (24 hr) than age- and gender-matched D21 control cells as measured by induced expression of the ISGs MX1, IDO1 and

*Figure 3 continued on next page*

*Figure 3 continued*

ISG15. Elevated pSTAT1 levels confirm effective induction of the IFN pathway in response to ligand exposure. (**B**) Western blots as in **A** for IFN-β treatment. (**C**) Western blots as in **A** for IFN-γ treatment. * indicates non-specific bands.

## Trisomy 21 cells display stronger induction of ISGs upon stimulation with IFN ligands than euploid cells

We next investigated whether T21 cells produce a stronger response to specific IFN ligands than their D21 counterparts. To test this, we treated three pairs of fibroblasts –roughly matched by age and sex– with various doses of the Type I ligands IFN-α or -β, or with the Type II ligand IFN-γ, and monitored the expression of key ISGs via western blot. We also monitored phosphorylation of STAT1. Overall, these efforts revealed that trisomy 21 cells show stronger induction of ISGs upon treatment with all three ligands, albeit with variation across specific cell lines and ligands (*Figure 3*). For example, stimulation with IFN-α led to stronger induction in the T21 cell line for MX1 in pairs 1 and 2, stronger induction of IDO1 in pairs 1 and 3, and stronger induction of ISG15 in pairs 1 and 2 (*Figure 3A*). Similar results were observed for the other Type I ligand, IFN-β. However, ligand-specific differences were also observed. For example, IDO1 was more strongly induced by IFN-α and -β in the T21 cell line in pair 1, but this was not the case when using IFN-γ (*Figure 3A–C*). Thus, these results confirm the notion of strong inter-individual variation in the downstream signaling effects of T21. Of note, all three IFN ligands consistently induced STAT1 phosphorylation (pSTAT1) both in D21 and T21 cells, but the levels of pSTAT1 did not correlate precisely with the expression levels of the various ISGs. For example, the obviously different levels of ISG15 in pair 2 upon treatment with the three ligands do not correlate with dissimilar levels of pSTAT1 (*Figure 3A–C*). This suggests that STAT1 phosphorylation is not a robust predictor of ISG expression, which is ultimately defined by the orchestrated action of multiple IFN-activated transcription factors.

## A kinome shRNA screen identifies the IFN-activated kinases JAK1 and TYK2 as negative regulators of cell viability in trisomy 21 fibroblasts

In a parallel unbiased approach to identify signaling cascades deregulated by T21, we employed an shRNA screen to identify protein kinases that may have a differential impact on the viability (i.e. proliferation and/or survival) of T21 cells relative to D21 cells. We hypothesized that core gene expression changes in T21 cells lead to a rewiring of signaling cascades, creating differential requirements for specific kinases during cell survival and proliferation. In order to identify such kinases, we introduced a library of 3,075 shRNAs targeting 654 kinases into each of the 12 fibroblast cell lines we subjected to transcriptome analysis. We then propagated these cells for 14 days to allow for selection of cells harboring shRNAs targeting kinases that differentially affect survival and/or proliferation of T21 cells versus D21 cells, henceforth referred to as DM^T21 kinases (Differential Modulators of T21 cells) (*Figure 4A*). In this screen, relative enrichment of a given shRNA in the T21 population could result from the targeted kinase being a negative regulator of T21 cellular fitness, a positive regulator of D21 cellular fitness, or a combination of both. To minimize the possibility of shRNA off-target effects, we required at least three independent shRNAs targeting a given kinase to score as significantly enriched or depleted, with no more than one shRNA against each kinase scoring in the opposite direction (see Materials and methods for details). This analysis identified a total of 25 and 15 kinases that negatively and positively affect the fitness of T21 cells relative to D21 cells, respectively (*Figure 4B*, *Figure 4—figure supplement 1*, *Supplementary file 2*). The top scoring enriched kinase was mTOR, indicating that this kinase differentially decreases the fitness of T21 cells (and/or differentially increases the fitness of D21 cells). This could be consistent with previous reports showing hyperactivation of mTOR signaling in the brains of individuals with DS and mouse models of trisomy 21 and consequent impairments in autophagy (*Ahmed et al., 2013*; *Perluigi et al., 2015*; *Troca-Marín et al., 2014*; *Iyer et al., 2014*). Importantly, among DM^T21 kinases predicted to hinder T21 cell viability were the IFN-activated kinases JAK1 and TYK2 (*Müller et al., 1993*; *Stahl et al., 1994*) (*Figure 4B,C*, *Figure 4—figure supplement 1A,B*). To confirm that JAK1 signaling negatively affects the relative viability of T21 cells, we treated two pairs of D21/T21 fibroblasts with increasing doses of the JAK1/2 inhibitor ruxolitinib (Rux) (*Tefferi et al., 2011*). Rux treatment led to decreased

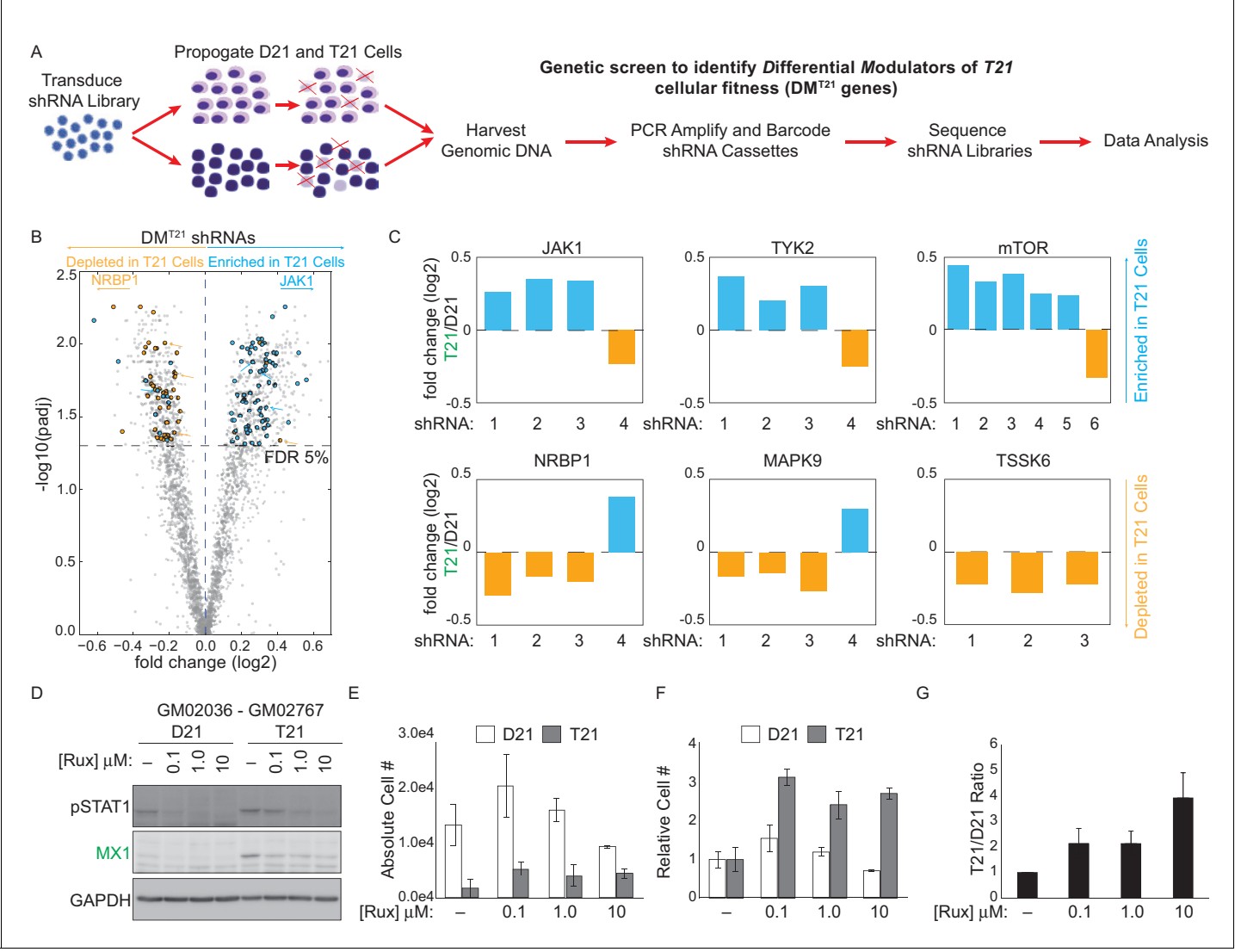

**Figure 4.** An shRNA screen identifies the interferon (IFN)-activated kinases JAK1 and TYK2 as negative regulators of trisomy 21 (T21) cellular fitness. (A) Schematic of kinome-focused shRNA screen to identify Differential Modulators of T21 (DM$^{T21}$) cellular fitness. (B) Volcano plot highlighting shRNAs targeting DM$^{T21}$ genes that differentially inhibit T21 (blue) or euploid (D21, yellow) cellular fitness. Top hits were filtered by a FDR < 5% and at least three shRNAs to the same gene scoring in one direction with no more than one shRNA scoring in the opposite direction. NRBP1 and JAK1 shRNAs are indicated with arrows. (C) Bar graphs of the screen results for the IFN-related kinases JAK1 and TYK2, as well as mTOR, NRBP1, MAPK9 and TSSK6. (D) Western blot analysis confirming downregulation of STAT1 phosphorylation and MX1 expression upon inhibition of JAK kinases with ruxolitinib (Rux) at the indicated concentrations in the GM02036-GM02767 cell pair. (E) Absolute cell numbers grown for 72 hr in their respective conditioned media with the indicated doses of Rux. (F) Relative cell numbers from (E). (G) Ratio of T21:D21 relative cell numbers demonstrates the overall differential effect of Rux on the number of viable cells from this T21-D21 pair. Results from a second cell line pair are shown in *Figure 4—figure supplement 1D–G*. All data shown are an average of three experiments ± standard error of the mean.

The following figure supplement is available for figure 4:

**Figure supplement 1.** An shRNA screen identifies differential modulators of T21 (DM$^{T21}$) cellular fitness.

levels of pSTAT1, decreased protein expression of MX1 –an ISG encoded on chr21–, and decreased mRNA expression of several ISGs found to be upregulated in T21 fibroblasts in our RNA-seq experiment (*Figure 4D* and *Figure 4—figure supplement 1C,D*). To assess the impact of Rux treatment on cell viability, we seeded equal numbers of D21 and T21 fibroblasts in the absence or presence of increasing doses of the inhibitor, and counted the number of viable cells 3 days post-seeding.

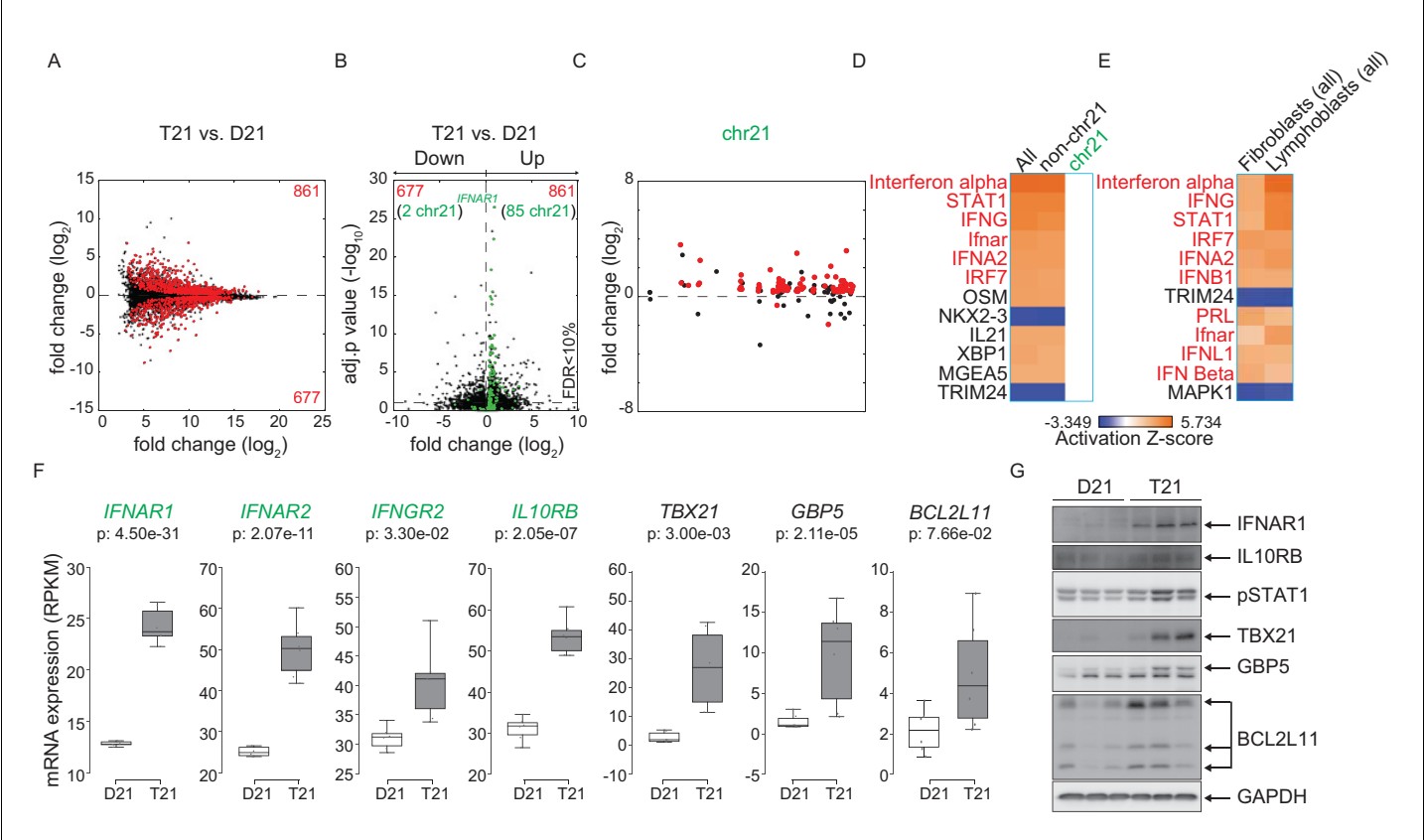

**Figure 5.** Activation of the interferon (IFN) transcriptional response is conserved in trisomy 21 (T21) lymphoblastoid cell lines. (**A**) MA plot displaying the gene expression signature associated with T21 in a panel of six lymphoblastoid cell lines, three of which harbor T21. Differential expressed genes (DEGs), as defined by DEseq2 (FDR < 10%), are labeled in red. (**B**) Volcano plot of DEGs with those encoded on chr21 highlighted in green. (**C**) Manhattan plot of chr21 with DEGs in red and all other genes in black. (**D**) Upstream regulator analysis reveals activation of the IFN transcriptional response in T21 lymphoblastoid cell lines. (**E**) Comparative analysis between fibroblasts and lymphoblastoids highlights conserved upstream regulators within the IFN pathway. (**F**) Box and whisker plots of RNA expression for the four IFN receptor subunits encoded on chr21 (green) and three interferon-related genes (black). mRNA expression values are displayed in reads per kilobase per million (RPKM). Benjamini-Hochberg adjusted p-values were calculated using DESeq2. (**G**) Western blot analysis confirming upregulation of IFN receptors, pSTAT1, and interferon related genes, at the protein level in T21 lymphoblastoids.

The following figure supplements are available for figure 5:

**Figure supplement 1.** Biological replicates of lymphoblastoid samples are highly related.

**Figure supplement 2.** Differentially expressed genes in trisomy 21 lymphoblastoid cell lines are not organized into obvious chromatin domains.

**Figure supplement 3.** Components of the IFN response are activated in a mouse model of Down syndrome.

---

Notably, the number of viable T21 cells was much lower in all conditions tested (*Figure 4E* and *Figure 4—figure supplement 1E*). However, whereas Rux treatment led to a dose-dependent increase in the number of viable T21 cells, it also produced a decrease in the number of viable D21 cells at the highest concentration. When the cell counts are represented as T21/D21 ratios, it is clear that JAK inhibition has a differential effect on cell proliferation between T21 and D21 cells (*Figure 4F,G* and *Figure 4—figure supplement 1F,G*). This is consistent with shRNAs targeting JAK1 (and TYK2) being differentially enriched in T21 cells during the 14-day course of the screen. Ultimately, these data support the notion of differential signaling requirements in T21 relative to D21 cells and identify two IFN-related kinases as negative regulators of T21 fibroblast viability.

## Activation of the IFN response by trisomy 21 is conserved in lymphoblastoid cells

To test whether consistent changes in gene expression programs elicited by trisomy 21 are conserved across cell types, we performed RNA-seq on a panel of six age-matched, female lymphoblastoid cell lines from D21 and T21 individuals (*Figure 5—figure supplement 1A–B*). These cell lines were generated by immortalizing B cells with Epstein Barr Virus (EBV), thus enabling us to compare a cell type of lymphocytic origin with the fibroblasts of mesenchymal origin. Analysis of DEGs associated with T21 identified 1538 genes both up and downregulated with more upregulated DEGs (861 out of 1538), as was seen in the fibroblasts (*Figure 5A*, *Supplementary file 1B*). Similarly, a peak of highly significant DEGs with ~1.5-fold change, comprised of chr21-encoded genes, is observed in a volcano plot (*Figure 5B*). Furthermore, most DEGs are distributed across the genome, and not arranged into obvious chromosomal domains outside of chr21 (*Figure 5C* and *Figure 5—figure supplement 2*). IPA revealed that the top upstream regulators of the consistent gene expression signature driven by T21 in lymphoblastoids are also IFN-related, and that this prediction is powered by non-chr21 DEGs (*Figure 5D*). Comparison of DEGs from fibroblasts and lymphoblastoids demonstrates that many of the same upstream regulators are predicted to be activated and are IFN-related factors (*Figure 5E*). All four chr21-encoded IFN receptors are significantly upregulated in lymphoblastoids (*Figure 5F*), as they are in fibroblasts. In fact, the most significant DEG encoded on chr21 is *IFNAR1* (*Figure 5B*). Increased basal protein expression was confirmed by western blot for IFNAR1 and IL10RB, as well as for the interferon-related genes TBX21, GBP5 and BCL2L11 (BIM) (*Figure 5G*). STAT1 phosphorylation was also elevated in the T21 lymphoblastoids (*Figure 5G*).

We next wanted to determine if the IFN signature was conserved in a mouse model of Down syndrome. Dp16 mice were selected because they contain a region of mouse chromosome 16 syntenic to human chromosome 21 that includes the IFN receptor cluster, without triplication of non-syntenic regions (*Li et al., 2007*). RNA-seq was performed on the LSK (Lineage negative, Sca1 positive, c-Kit positive) population of multipotent hematopoietic stem and progenitor cells obtained from the bone marrow of Dp16 mice and matched littermate controls. These results confirmed that three of the four IFN receptors are upregulated in Dp16 mice (*Ifnar1*, *Ifnar2*, and *Ifngr2*), along with several canonical ISGs (*Figure 5—figure supplement 3*, *Supplementary file 1C*). Our results demonstrate that IFN activation by trisomy 21 is conserved in the hematopoietic lineage.

## The IFN response is activated in circulating blood cell types of individuals with trisomy 21

In order to determine whether our findings are applicable to living human individuals with T21, we isolated monocytes, T cells, and B cells, from 10 individuals with T21 and seven D21 individuals. As for our cell line work, we included samples from both sexes with varying ages and genetic backgrounds (*Figure 6—figure supplement 1A,B*). Monocytes and T cells were subjected to transcriptome analysis by RNA-seq, and B cells used for IFN receptor surface expression analysis by flow cytometry. The transcriptome analyses identified hundreds of consistent gene expression changes associated with T21 in both cell types, with the expected ~1.5x fold increase in chr21 gene expression (*Figure 6—figure supplement 1C,D*). The IFN receptors encoded on chr21 are significantly upregulated in circulating blood cell types from individuals with T21, with the sole exception of *IFNGR2* in T cells (*Figure 6A,B*, *Supplementary file 1D*). Flow cytometry detected a minor increase in surface expression of IFNAR1, IFNGR2, and IL10RB, in the B cell population, but not for IFNGR1, which is not encoded on chr21 (*Figure 6—figure supplement 2*). Once again, upstream regulator analysis identified IFN ligands and IFN-activated transcription factors as predicted drivers of gene induction in T21 monocytes and T cells (*Figure 6C* and *Figure 6—figure supplement 3*) with many canonical ISGs scoring among the most significantly induced genes (*Figure 6A,B*).

A comparison of the upstream regulator analyses of the four cell types included in this study revealed both conserved and cell type-specific features. The upstream regulator analysis shows that IFN activation is conserved, as is predicted inactivation of the IFN repressors MAPK1 and TRIM24 (*Figure 6C*). However, a unique feature of the primary cell types -monocytes and T cells- is a predicted inactivation of the gene expression program driven by the transcription factor MYCN (*Figure 6C*). Comparison of the canonical pathways deregulated in all four cell types confirms that IFN signaling is the top activated pathway, but also reveals that monocytes and T cells, and to a

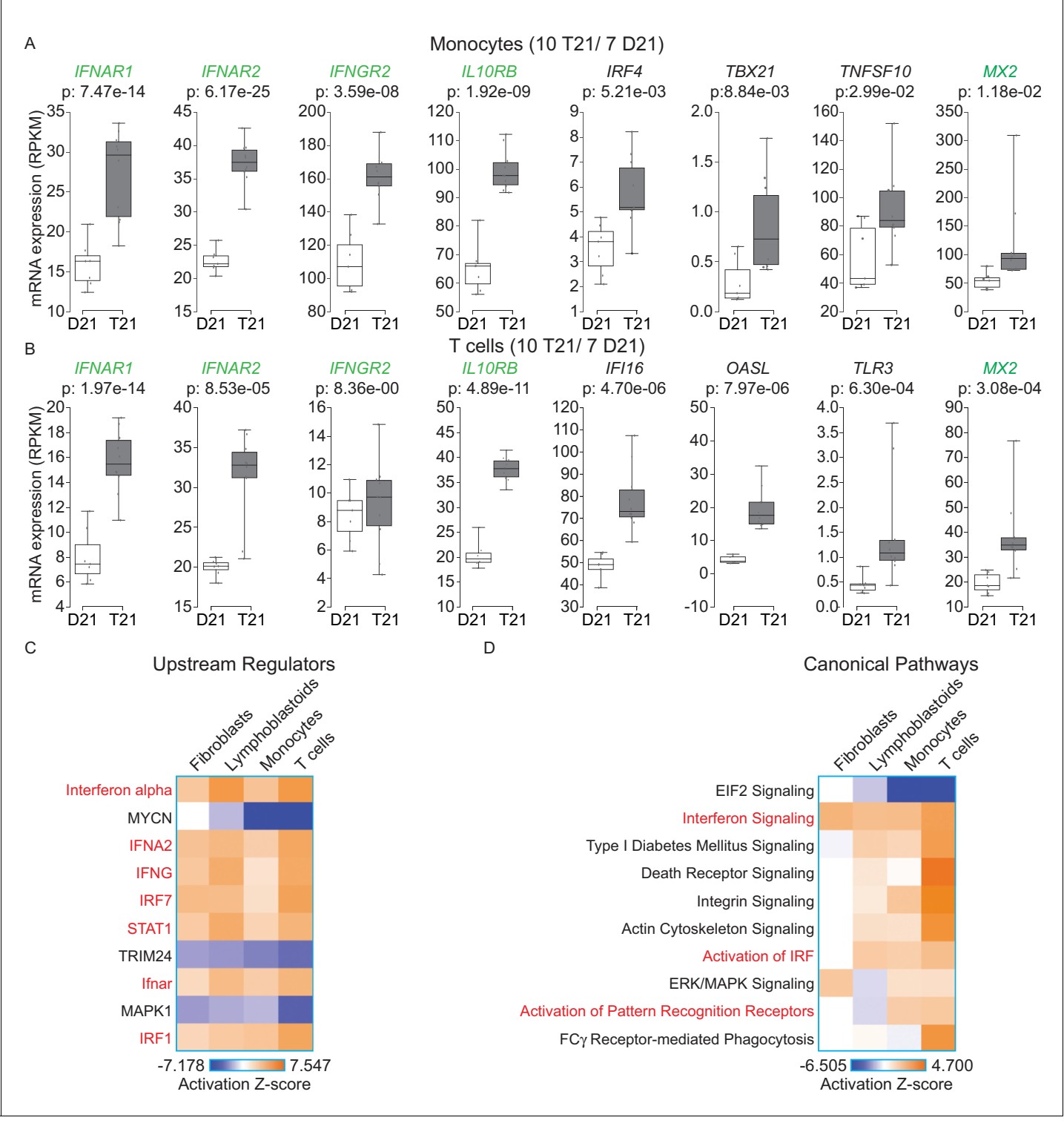

**Figure 6.** IFN signaling is activated in circulating blood cells from individuals with T21. (**A**) Box and whisker plots of RNA expression for the four IFN receptor subunits encoded on chr21 and representative IFN-related genes in circulating monocytes. mRNA expression values are displayed in reads per kilobase per million (RPKM). Benjamini-Hochberg adjusted p-values were calculated using DESeq2. (**B**) Box and whisker plots of RNA expression as in (A) for circulating T cells. (**C**) Upstream regulator analysis reveals activation of the IFN transcriptional response in T21 monocytes and T cells, as well as downregulation of the MYCN-driven transcriptional program. (**D**) Canonical pathway analysis reveals activation of the IFN signaling pathway in T21 monocytes and T cells, as well as downregulation of the EIF2 signaling pathway.

*Figure 6 continued on next page*

*Figure 6 continued*

The following figure supplements are available for figure 6:

**Figure supplement 1.** Effects of T21 on the transcriptome of circulating monocytes and T cells from individuals with T21 and typical controls.
**Figure supplement 2.** Surface expression of IFN receptors is increased in B cells from individuals with T21.
**Figure supplement 3.** The IFN gene signature from monocytes and T cells is largely encoded by non-chr21 genes.

lesser degree lymphoblastoids, show strong repression of the EIF2 pathway (*Figure 6D*). Since both MYCN and EIF2 are potent regulators of protein synthesis, we decided to investigate this observation in more detail.

## Trisomy 21 downregulates the translation machinery in primary monocytes and T cells

A well-established aspect of the IFN response is the selective control of protein translation, purportedly to prevent the synthesis of viral proteins during the course of infection (*Johnson et al., 1968*). Mechanistically, it has been shown that IFN signaling impairs processing of rRNAs and controls the

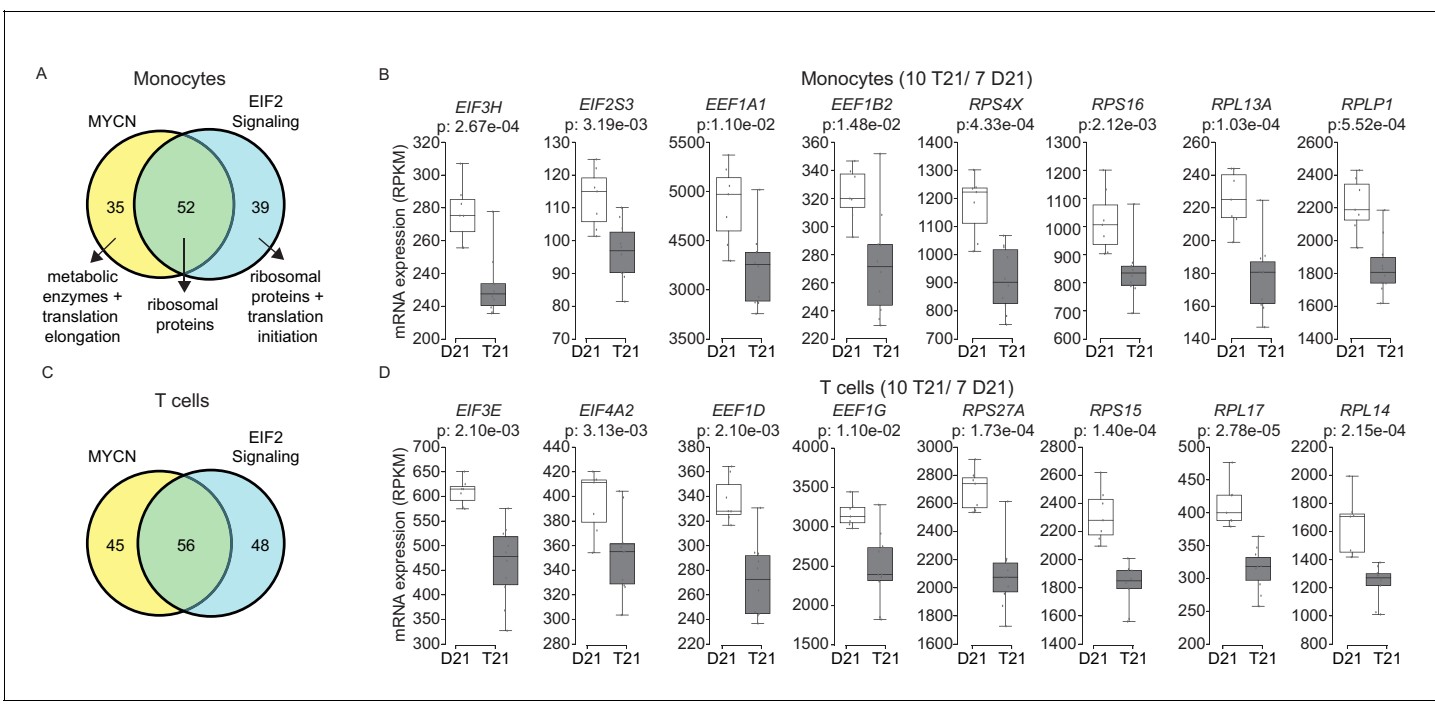

**Figure 7.** Trisomy 21 globally downregulates the translation machinery in monocytes and T cells. (**A**) Venn diagram demonstrating the overlap in DEGs comprising the MYCN upstream regulator and EIF2 signaling pathway gene signatures identified by IPA in monocytes. Prominent components of each group are indicated with arrows. See also *Figure 7—figure supplement 2*. (**B**) Box and whisker plots of RNA expression for representative translation-related genes from monocytes. mRNA expression values are displayed in reads per kilobase per million (RPKM). Benjamini-Hochberg adjusted p-values were calculated using DESeq2. (**C**) Venn diagram demonstrating the overlap in DEGs as in (**A**) for T cells. (**D**) Box and whisker plots of RNA expression as in (**C**) for T cells.

The following figure supplements are available for figure 7:

**Figure supplement 1.** The MYCN transcriptional program is downregulated by T21.
**Figure supplement 2.** The EIF2 Signaling pathway is downregulated by T21.

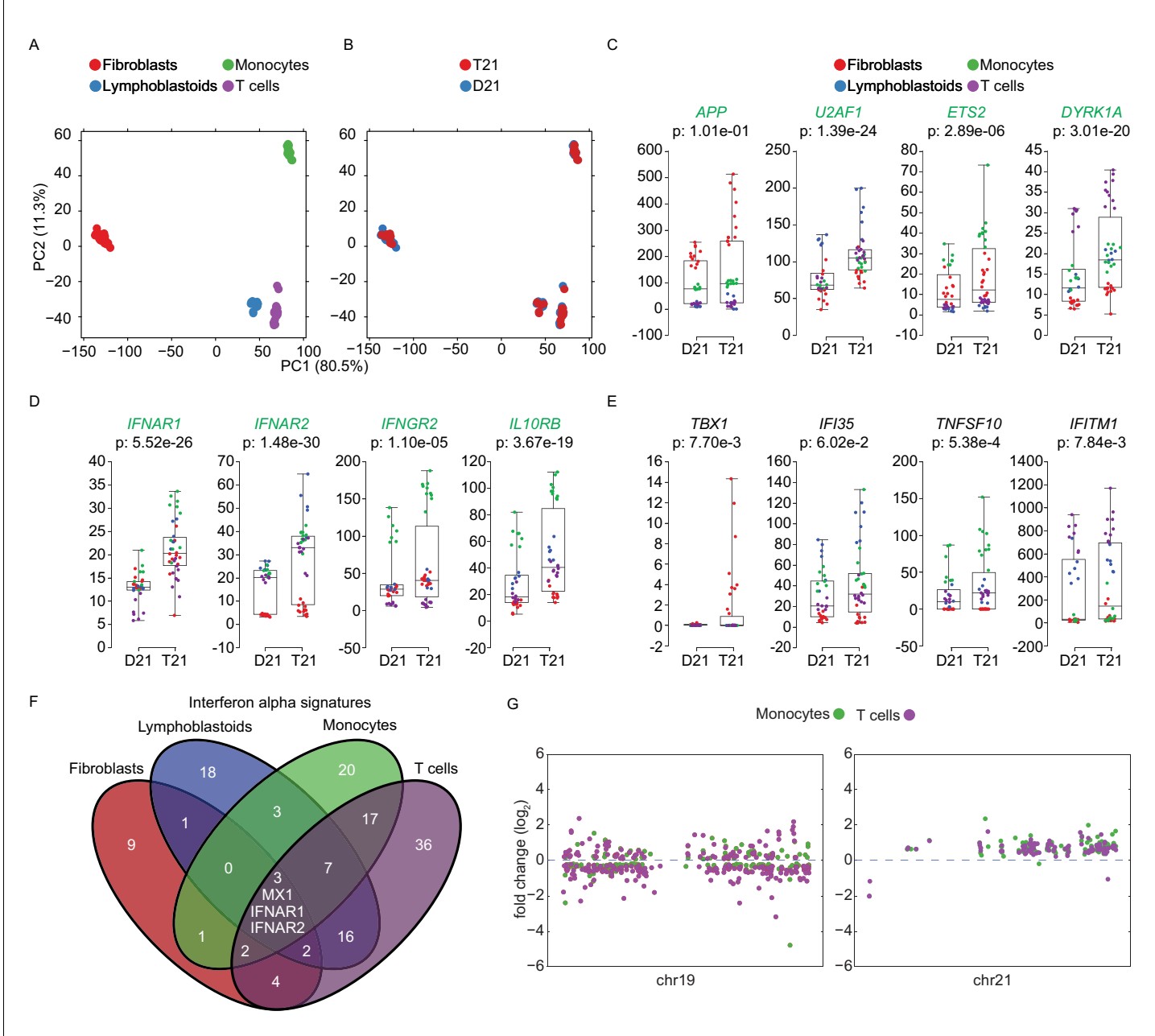

**Figure 8.** Trisomy 21 activates the IFN gene expression program in a cell type-specific manner. (**A**) Principal component analysis (PCA) of all RNA-seq samples from this study colored by cell type. (**B**) PCA analysis as in (**A**) colored by chr21 copy number. (**C**) Box and whisker plots of RNA expression for representative chr21-encoded genes from all samples. mRNA expression values are displayed in reads per kilobase per million (RPKM). Benjamini-Hochberg adjusted p-values were calculated using DESeq2 by comparing all T21 samples to all D21 samples. Individual data points are colored by cell type. (**D,E**) Box and whisker plots as in (**C**) for chr21-encoded IFN receptors and representative ISGs. (**F**) Venn diagram showing the cell type-specificity of the *Interferon alpha* gene expression programs identified by IPA for each cell type. (**G**) Manhattan plots for chromosomes 19 and 21 comparing the DEGs from monocytes and T cells derived from the same individuals.

activity and/or expression of specific translation factors (*Walsh et al., 2013*; *Maroun, 1978*). On the other hand, the MYC family of transcription factors are known drivers of ribosome biogenesis, protein synthesis and cell growth (*van Riggelen et al., 2010*; *Boon et al., 2001*; *Kim et al., 2000*; *Arabi et al., 2005*). Similarly, the EIF2 pathway is a key driver of protein translation, with eIF2 itself being an essential translation initiation factor (*Hinnebusch, 2014*). Analysis of the gene signatures

identified by IPA that predicted inactivation of both the MYCN transcriptional program and the EIF2 pathway showed a substantial degree of overlap (*Figure 7A,C*, *Supplementary file 1E*). In monocytes and T cells, the genes common between the two repressed programs encode components of both the small and large ribosome subunits (i.e. RPS proteins in the 40S complex and RPL proteins in the 60S complex) (*Figure 7A,C*, *Figure 7—figure supplements 1* and *2*). Genes exclusive to the MYCN signature are enriched for metabolic enzymes and translation elongation factors (EEFs). Genes exclusive to the EIF2 signature are enriched for translation initiation factors (EIFs) and additional ribosomal proteins. Examples of RPSs, RPLs, EEFs and EIFs downregulated in trisomy 21 cells are shown in *Figure 7B and D* (see also *Figure 7—figure supplements 1* and *2*). This result is consistent with reports that interferon treatment results in a global decrease in expression of the translation machinery in primary PBMCs (*Taylor et al., 2007*; *Gupta et al., 2012*). Altogether, these results indicate that T21 causes a general downregulation of dozens of components of the protein synthesis machinery in circulating monocytes and T cells.

## Trisomy 21 elicits cell type-specific IFN transcriptional responses

Having performed transcriptome analysis of cell types of different origins, we investigated to what degree the gene expression changes caused by T21 are affected by cell type-specific regulatory landscapes. A principal component analysis (PCA) shows the fibroblast transcriptomes segregating strongly (PC1 80.5%) from those of the cell types of hematopoietic origin (*Figure 8A*). B cell-derived lymphoblastoids and T lymphocytes cluster together, yet they segregate away from the monocytes of myeloid origin (PC2, 11.3%). Within this background, the global impact of the trisomy on the transcriptome is secondary to the effects of the cell type of origin (*Figure 8B*). Next, we asked to what degree genes encoded on chr21 could be affected by these cell type-specific regulatory landscapes. Indeed, it was easy to identify many chr21 genes displaying obvious differences in relative expression among cell types. For example, *APP* is relatively more highly expressed in fibroblasts, *U2AF1* more highly expressed in lymphoblastoids, *ETS2* more highly expressed in monocytes, and *DYRK1A* more highly expressed in T cells (*Figure 8C*, *Supplementary file 1F*). The IFN receptors on chr21 also showed some degree of cell type-specific expression (e.g. *IFNAR2* lowly expressed in fibroblasts, *IFNGR2* lowly expressed in T cells, *Figure 8D*). Furthermore, relative differences in cell type-specific expression is also evident for canonical ISGs (*Figure 8E*). These observations led us to ask to what degree the IFN transcriptional response elicited by T21 is conserved across cell types. To address this, we compared the DEGs comprising the T21-induced *Interferon alpha* signature identified by IPA in each cell type (*Figure 6C*). Remarkably, this exercise revealed a large degree of cell type-specificity, with most IFN-related genes being differentially expressed in only one cell type (*Figure 8F*). In fact, the only common genes among all four signatures are three IFN-α-related genes encoded on chr21: *IFNAR1*, *IFNAR2*, and *MX1*. Expectedly, lymphoblastoids and T cells showed a greater degree of overlap than other pairwise comparisons. Overall, these results indicate that while T21 operates within, and is modulated by, cell type-specific regulatory landscapes, it nonetheless activates the IFN transcriptional response consistently by inducing different gene sets within this program. This is in stark contrast to the notion that T21 affects gene expression either stochastically or through large rearrangements of chromatin domains. In fact, Manhattan plots of the DEGs in monocytes and T cells derived from the same individuals not only confirm the absence of large domains of chromatin deregulation, but also highlight the high degree of cell type-specific changes caused by the trisomy (*Figure 8G*).

## Discussion

We report here that T21 leads to consistent activation of the IFN pathway. As discussed below, IFN hyperactivation could explain many of the developmental and clinical impacts of T21. In fact, we posit that Down syndrome can be understood largely as an interferonopathy, and that the variable clinical manifestations of T21 could be explained by inter-individual differences in adaptation to chronic IFN hyperactivity.

The link between IFN signaling and T21 is not entirely unprecedented. More than 40 years ago, it was found that human T21 fibroblasts, but not those trisomic for chr13 or chr18, have increased sensitivity to IFN exposure and are more resistant to viral infection (*Tan et al., 1974a*, *1974b*). In fact, somatic cell hybrid experiments showed that chr21 is sufficient to confer sensitivity to human IFN in

mouse cells (*Slate et al., 1978*). Pioneering work by Maroun and colleagues using an early mouse model of DS carrying an extra copy of chr16 that harbors orthologues of many human chr21 genes, including the four IFN receptors, clearly implicated IFN as a contributor to the deleterious effects of the trisomy. For example, treatment of pregnant female mice with anti-IFN antibodies resulted in the partial rescue of embryonic growth defects and embryonic lethality (*Maroun, 1995*). Furthermore, partial normalization of gene dosage for the IFN receptor subunits via gene knockout was shown to improve embryonic development and survival of T21 cortical neurons in vitro (*Maroun et al., 2000*). More recently, a study found global disruption of IFN-related gene networks in the brains of the Ts1Cje mouse model of DS, which also carries triplication of the IFN receptor subunits (*Ling et al., 2014*). However, deeper investigations of IFN signaling in human T21 cells and tissues are largely absent from the literature of the past 30 years, with a few exceptions, such as the description of IFN signaling as a contributor to periodontal disease in DS (*Tanaka et al., 2012*; *Iwamoto et al., 2009*). Collectively, these reports and the genomics analyses reported here demonstrate that activation of the IFN pathway in T21 cells is a widespread phenomenon that occurs in diverse tissues, and that is relevant to human Down syndrome as well as the various mouse models of DS with triplication of IFN receptors.

Constitutive activation of IFN signaling could conceivably explain a large number of comorbidities associated with DS, such as the increased risk of transient myeloproliferative disorder, diverse leukemias, several autoimmune disorders (*Richardson et al., 2011*), and perhaps even the lower rate of solid tumors (*Zitvogel et al., 2015*; *Hasle et al., 2016*). Importantly, several JAK inhibitors are either approved or being tested in clinical trials for the treatment of several conditions associated with DS –albeit in the typical population–, including myeloproliferative, inflammatory and autoimmune disorders, as well as leukemia (*Padron et al., 2016*; *Spaner et al., 2016*; *Tefferi et al., 2011*; *Quintás-Cardama et al., 2010*; *Shi et al., 2014*; *Keystone et al., 2015*; *Jabbari et al., 2015*). It should be noted, however, that the dose limiting toxicities of JAK inhibitors, like ruxolitinib, are anemia and thrombocytopenia (*McKeage, 2015*; *Plosker, 2015*). Therefore, rigorous clinical investigations will be required to define if there is a therapeutic window in which these drugs would benefit individuals with DS before the appearance of toxicity. Additional research will also be required to elucidate the interplay between hyperactive IFN signaling in DS with other important factors encoded on chr21 (e.g. *DYRK1A, APP*) (*Malinge et al., 2012*; *Wiseman et al., 2015*) or elsewhere in the genome, that have been involved in the development of the specific comorbidities. For example, the Sonic Hedgehog (SHH) pathway has been implicated in the etiology of structural and cognitive defects in a mouse model of DS, including cerebellar atrophy (*Das et al., 2013*). Interestingly, IFN signaling has been shown to crosstalk with the SHH pathway, and cerebellar atrophy is also a hallmark of Type I Interferonopathies (*Moisan et al., 2015*; *Sun et al., 2010*; *McGlasson et al., 2015*; *Crow and Manel, 2015*).

Increased JAK/STAT signaling has been postulated to contribute to some of the neurological features of DS (*Lee et al., 2016*). Notably, it has been reported that therapeutic exposure to interferons can produce diverse types of neurological dysfunction, including depression, cerebral palsy and spastic diplegia (*Wichers et al., 2005*; *Grether et al., 1999*; *Wörle et al., 1999*; *Barlow et al., 1998*). Furthermore, a large number of neurological conditions have been linked to deregulated IFN signaling, most prominent among them the so called Type I Interferonopathies (*McGlasson et al., 2015*; *Crow and Manel, 2015*). Therefore, we propose that constitutive activation of the IFN pathway in the central nervous system of individuals with DS is responsible for many of the neurological problems caused by the trisomy. In particular, IFN-mediated activation of microglia could lead to neurotoxicity by several mechanisms, including serotonin depletion, generation of reactive oxygen species, and excitatory toxicity, which could potentially be ameliorated with inhibitors of the IDO1 enzyme, a key ISG (*Wichers and Maes, 2004*; *Wichers et al., 2005*). Although much research remains to be done, it is now possible to envision early intervention strategies to ameliorate the variable ill effects of T21 by using pharmacological inhibitors of the IFN pathway.

# Materials and methods

## Cell culture and drug treatments

Six human fibroblast lines from individuals with trisomy 21 (T21) and six approximately age- and sex-matched fibroblast lines from typical individuals (D21) were obtained from the Coriell Cell Repository (Camden, NJ) and immortalized with hTERT as described (*Lindvall et al., 2003*). EBV-immortalized lymphoblastoid lines, three T21 and three D21, were obtained from the Nexus Clinical Data Registry and Biobank at the University of Colorado. Fibroblasts were maintained in DMEM and lymphoblastoids were maintained in RPMI medium in a humidified 5% $CO_2$ incubator at 37°C. The media was supplemented with 10% fetal bovine serum and 1% antibiotic/antimycotic and was changed every 3–6 days. Fibroblast monolayers were serially passaged by trypsin-EDTA treatment, and lymphoblastoids were serially passaged via dilution in fresh media. Fibroblast lines used in this study are described in *Figure 1—figure supplement 1*. All cell lines were confirmed mycoplasma negative by PCR as previously described (*Uphoff and Drexler, 2002*). T21 status was authenticated as described in *Figure 1—figure supplement 1D*. Research Resource Identifiers (RRIDs) for fibroblast cell lines are:

| Line | RRID # |
| --- | --- |
| GM08447 | CVCL_7487 |
| GM05659 | CVCL_7434 |
| GM00969 | CVCL_7311 |
| GM02036 | CVCL_7348 |
| GM03377 | CVCL_7384 |
| GM03440 | CVCL_7388 |
| GM04616 | CVCL_V475 |
| AG05397 | CVCL_L780 |
| AG06922 | CVCL_X793 |
| GM02767 | CVCL_V469 |
| AG08941 | CVCL_X871 |
| AG08942 | CVCL_X872 |

## Interferon treatment in cell culture

Recombinant human interferons alpha 2A (11101–2, R&D Systems), beta (300-02BC, Peprotech) and gamma (PHC4031, Gibco) were obtained from Thermo Fisher Scientific (Waltham, MA), aliquoted, and stored at −80°C. Three T21 fibroblast lines and their age- and sex-matched D21 fibroblast counterparts were plated at equivalent densities and grown 72 hr to ensure similar cycling of the cells, then re-plated at equivalent densities and incubated overnight. Media was removed the following day and replaced with media containing the indicated doses of interferon ligands dissolved in PBS or vehicle (PBS alone). All media were normalized for final PBS concentration at highest interferon dose. Cells were grown an additional 24 hr after interferon application, then the media removed, cells washed with PBS and harvested via cell scraping. The harvested cells were pelleted and lysed in RIPA buffer with protease and phosphatase inhibitors.

## JAK inhibition in cell culture

Ruxolitinib (INCB018424) was obtained from Selleck Chemicals (Houston, TX, S1378) and dissolved in DMSO to make a 5 mM stock solution and stored at −20°C. Fibroblast lines were plated at equivalent cell numbers and allowed to grow for 72 hr in order to condition the media with secreted factors. The conditioned media was harvested and stored at 4°C for 3–7 days prior to use. One T21 fibroblast line and its age- and sex-matched D21 fibroblast counterpart were plated at equivalent cell numbers in their respective conditioned media and incubated overnight. Plating media was removed the following day and replaced with conditioned media containing the indicated doses of

ruxolitinib or DMSO. All conditioned drug media was normalized for DMSO concentration. Cells were grown an additional 72 hr after drug application, harvested with trypsin-EDTA, and counted with 0.2% trypan blue using a hemocytometer.

## Western blots

Cells were plated at equal densities and allowed to grow 72 hr before harvesting cell pellets. Pellets were washed with PBS and resuspended in RIPA buffer containing 1 µg/mL pepstatin, 2 µg/mL aprotonin, 20 µg/mL trypsin inhibitor, 10 nM leupeptin, 200 nM $Na_3VO_4$, 500 nM phenylmethylsulfonyl fluoride (PMSF), and 10 µM NaF. Suspensions were sonicated at six watts for 15 s two times and clarified by centrifugation at 21,000 g for 30 min at 4°C. Supernatants were quantified in a Pierce BCA Protein Assay and diluted in complete RIPA with 4x Laemmli sample buffer. Tris-glycine SDS-polyacrylamide gel electrophoresis was used to separate 20–40 µg protein lysate, which was transferred to a 0.2 µm polyvinylidene fluoride (PVDF) membrane. Membranes were blocked in 5% non-fat dried milk or 5% bovine serum albumin (BSA) in Tris-buffered saline containing 0.1% TWEEN (TBS-T) at room temperature for 30–60 min before probing overnight with primary antibody in 5% non-fat dried milk or 5% BSA in TBS-T at 4°C while shaking. Membranes were washed 3x in TBS-T for 5–15 min before probing with a horseradish peroxidase (HRP) conjugated secondary antibody in 5% non-fat dried milk or 5% BSA at room temperature for one hour. Membranes were again washed 3x in TBS-T for 5–15 min before applying enhanced chemiluminescence (ECL) solution. Chemiluminensense signal was captured using a GE (Pittsburgh, PA) ImageQuant LAS4000.

Antibodies used in this study:

| Antibody | Manufacturer | Product # | RRID # |
|---|---|---|---|
| anti-mouse IgG-HRP | Santa Cruz Biotechnology (Dallas, TX) | sc-2005 | AB_631736 |
| anti-rabbit IgG-HRP | Santa Cruz Biotechnology | sc-2317 | AB_641182 |
| BIM | Cell Signaling Technology (Danvers, MA) | 2819 | AB_659953 |
| GAPDH | Santa Cruz Biotechnology | sc-365062 | AB_10847862 |
| GBP5 | Abcam (Cambridge, MA) | ab96119 | AB_10678091 |
| IFI27 | Abcam | ab171919 | N/A |
| IFNAR1 | R&D Systems | AF245 | AB_355270 |
| IFNGR2 | R&D Systems | AF773 | AB_355589 |
| IL10RB | R&D Systems | AF874 | AB_355677 |
| ISG15 | Cell Signaling Technology | 2743 | AB_2126201 |
| MX1 | Abcam | ab95926 | AB_10677452 |
| pSTAT1 | Cell Signaling Technology | 7649 | AB_10950970 |
| TBX21 | Cell Signaling Technology | 5214 | AB_10692112 |

## Q-RT-PCR

Total RNA was isolated using Trizol (Thermo Fisher) according to manufacturer's instructions. cDNA was synthesized using the qScript kit from Quanta Biosciences (Beverly, MA). PCR was performed using SYBR Select on a Viia7 from Life Technologies (Thermo Fisher).

Oligonucleotides used in this study:

| Gene ID | Accession # | Forward | Reverse |
|---|---|---|---|
| IFI27 | NM_001130080 | TCTGCAGTCACTGGGAGCAACT | AACCTCGCAATGACAGCCGCAA |
| IFITM1 | NM_003641 | TTCGCTCCACGCAGAAAACCA | ACAGCCACCTCATGTTCCTCCT |
| MX1 | NM_001144925 | TCCACAGAACCGCCAAGTCCAA | ATCTGGAAGTGGAGGCGGATCA |

| MX2 | NM_002463 | TCGGACTGCAGATCAAGGCTCT | CGTGGTGGCAATGTCCACGTTA |
| OAS1 | NM_001032409 | CCGCATGCAAATCAACCATGCC | TTGCCTGAGGAGCCACCCTTTA |
| OAS2 | NM_001032731 | AGGTGGCTCCTATGGACGGAAA | CGAGGATGTCACGTTGGCTTCT |

## RNA-seq from cell lines

Biological replicates for each cell line were obtained by independently growing cells in duplicate. Total RNA was purified from ~1 × $10^7$ logarithmically growing cells using Qiagen (Valencia, CA) RNeasy columns per manufacturer's instructions including on-column DNAse digestion. RNAs were quantified using a Take3 Micro-Volume plate in a Biotek (Winooski, VT) Synergy2 plate reader and their integrity confirmed using the Agilent RNA 6000 Pico Kit and the Agilent (Santa Clara, CA) 2100 Bioanalyzer System. 500 ng of total RNA with an RNA Integrity Number (RIN) greater than 7 were used to prepare sequencing libraries with the Illumina (San Diego, CA) TruSeq Stranded mRNA Library Prep Kit. Libraries were sequenced with an Illumina HiSeq 2000 System at the UCCC Genomics Core.

## Isolation of monocytes and T cells by fluorescence activated cell sorting (FACS)

Peripheral blood was collected in EDTA vacutainer tubes from 10 individuals with T21 and seven D21 controls. Blood was centrifuged at 500 g for 15 min to separate plasma, buffy coat and red blood cells (RBCs). Peripheral Blood Mononuclear Cells (PBMCs) were isolated from the buffy coat fraction by RBC lysis and 1x PBS wash according to manufacturer's instructions (BD, 555899). After RBC lysis and PBS wash, PBMCs were stained for sorts at 10–20 × $10^7$ cells/ml then diluted to approximately 5 × $10^7$ cells/ml in flow cytometry sorting buffer (1x PBS, 1 mM EDTA, 25 mM HEPES pH 7.0, 1% FBS). All staining was performed in flow cytometry sorting buffer with fluorochrome-conjugated antibodies for at least 15 min on ice while protected from light. Single cell suspensions were stained with CD45 (eBioscience, San Diego, CA, HI30, RRID:AB_467273), CD14 (Biolegend, San Diego, CA, 63D3, RRID:AB_2571928), CD3 (Biolegend, OKT3, RRID:AB_571907), CD16 (Biolegend, B73.1, RRID:AB_2616914), CD19 (Biolegend, HIB19, RRID:AB_2073119), CD56 (Biolegend, 5.1H11, RRID:AB_2565855) and CD34 (Biolegend, 561, RRID:AB_343601) antibodies. CD45+CD14+CD19-CD3-CD56- Monocytes and CD45+CD3+CD14-CD19-CD56- T cells were FAC-sorted into Dulbecco's Modified Eagle Medium (DMEM) supplemented with 4.5 g/L D-Glucose, L-Glutamine, and 5% FBS, on the MoFlo Astrios (Beckman Coulter, Brea, CA) at the CU-SOM Cancer Center Flow Cytometry Shared Resource.

## RNA extraction from monocytes and T cells

FAC-sorted cells were centrifuged at 500 g for 5 min and the media removed. Cells were resuspended in 350 µl RLT plus (Qiagen) and Beta-mercaptoethanol (BME) lysis buffer (10 µL BME:1 mL RLT plus) for downstream RNA isolation. Lysed cells were immediately stored at −80°C and RNA was later extracted using the AllPrep DNA/RNA/Protein Mini Kit according to manufacturer's instructions (Qiagen, 80004). RNA quality was determined by BioAnalyzer (Agilent) and quantified by Qubit (Life Technologies). Samples with RIN of 7 or greater and a minimum of 500 ng total RNA were used for library prep and sequencing.

## RNA-seq data analysis

Analysis of library complexity and high per-base sequence quality across all reads (i.e. q > 30) was performed using FastQC (v0.11.2) software (*Andrews, 2010*). Low quality bases (q < 10) were trimmed from the 3' end of reads and short reads (<30 nt after trimming) and adaptor sequences were removed using the fastqc-mcf tool from ea-utils. Common sources of sequence contamination such as mycoplasma, mitochondria, ribosomal RNA were identified and removed using FASTQ Screen (v0.4.4). Reads were aligned to GRCh37/hg19 using TopHat2 (v2.0.13, –b2-sensitive –keep-fasta-order –no-coverage-search –max-multihits 10 –library-type fr-firststrand) (*Kim et al., 2013*). High quality mapped reads (MAPQ > 10) were filtered with SAMtools (v0.1.19) (*Li et al., 2009*). Reads were sorted with Picardtools (SortSAM) and duplicates marked (MarkDuplicates). QC of final

reads was performed using RSeQC (v2.6) (*Wang et al., 2012*). Gene level counts were obtained using HTSeq (v0.6.1,–stranded=reverse –minaqual=10 –type=exon –idattr=gene –mode= intersection-nonempty, GTF-ftp://igenome:G3nom3s4u@ussd-ftp.illumina.com/Homo_sapiens/UCSC/hg19/Homo_sapiens_UCSC_hg19.tar.gz) (*Anders et al., 2015*). Differential expression was determined using DESeq2 (v1.6.3) and R (3.10) (*Love et al., 2014*). Volcano plots, manhattan plots, and violin plots, were made using the Python plotting library 'matplotlib' (http://matplotlib.org).

## shRNA screening

A pool of plasmids encoding 3,075 shRNAs targeting 654 kinases (kinome library) in the pLKO.1 backbone produced by The RNAi Consortium (TRC, Sigma-Aldrich, St. Louis, MO) were obtained from the University of Colorado Cancer Center Functional Genomics Shared Resource, as were the pΔ8.9 and pCMV-VSV-G lentiviral packaging plasmids. 2 μg of kinome library plasmid DNA at 100 ng/μL was mixed with 2 μg of packaging plasmid mix (at a 9:1 ratio of pΔ8.9:pCMV-VSV-G) at 100 ng/μL and incubated with 12 μg of Polyethylenimine for 15 min at RT. The entire mixture was then added to $3 \times 10^5$ HEK293FT packaging cells to give 100X coverage. 16 hr after transfection, media on cells was replaced with complete DMEM. 24 hr after media replacement, target cells were seeded at $1 \times 10^5$ cells/ well in a 6-well plate. Three wells for each line were combined at the time of harvest to reach a starting number of $3 \times 10^5$ cells per condition (again 100X coverage of the kinome library). 24 hr after seeding, the media from each well of packaging cells (now containing lentiviral library particles) was filtered through 0.45 μm cellulose acetate filters, diluted 1:3 into 6 mL of DMEM, and mixed with 6 μL of 8 mg/mL polybrene to facilitate transduction. This mixture was then used to transduce 3 wells (one total replicate) of each target cell line. 24 hr after transduction viral transduction, the media was replaced with fresh media. Finally, after an additional 24 hr, selection began by adding fresh DMEM with 1 μg/mL puromycin. Cells were then propagated for 14 days and genomic DNA harvested from all remaining cells using the Qiagen DNeasy Blood and Tissue kit with the optional RNAse A treatment step. Genomic DNA was quantified by $A_{260}$ using a Take3 micro-volume plate on a Synergy2 Microplate Reader. The quality of the genomic DNA was confirmed via electrophoresis on a 0.5% TAE agarose gel. Screens were performed in three independent biological replicates for each of the 12 fibroblast cell lines.

## shRNA library preparation

The library preparation strategy uses genomic DNA and two rounds of PCR in order to isolate the shRNA cassette and prepare a single strand of the hairpin for sequencing by means of an XhoI restriction digest in the stem loop region. This is critical as the hairpin secondary structures of shRNAs are not amenable to NGS and the TRC shRNAs do not have a long enough loop to allow PCR amplification of one shRNA arm in a single step. The first step in sequencing library preparation is to calculate how much genomic DNA must be used for PCR1 which isolates and amplifies the shRNA cassettes from genomic DNA using Phusion Polymerase. The oligonucleotides for PCR1 anneal to regions inside of the LTRs that are common to all clones in the library and should, therefore, amplify all shRNA cassettes with equal efficiency. Each reaction mixture for PCR1 consisted of 10 μL 5X Phusion *HF* buffer, 1 μL dNTPs (10 mM each), 2.5 μL pLKO Forward and Reverse primers (10 μM), 1 μL of 2 unit/μl Phusion Polymerase, 500 ng genomic DNA, and dH2O to 50 μL. The cycling conditions were as follows: 1 cycle of 98°C for 5 min, 15 to 25 cycles of 98°C for 30 s, 70°C for 30 s, 72°C for 30 s, and 1 cycle of 72°C for 7 min. 5 μL of each PCR1 were run on a 2% TAE agarose gel in order to visualize the expected band of 497 bp. It should be noted that the optimal PCR1 cycle number must be empirically determined for each library and to limit cycle numbers to minimize the effects of amplification bias. The correct product of PCR1 is 497 bp; however, excessive cycle numbers can result in the appearance of a slower migrating band. This band represents an annealing event between two amplification products with different shRNA sequences. As the majority of the 497 bp amplicon is common to all products, denatured PCR products can anneal to one another when not out-competed by an excess of primer in later cycles. This aberrant product does not correctly anneal within the central shRNA-containing sequence, therefore disrupting the double-stranded XhoI site required for the subsequent restriction digestion. Carefully determining the appropriate number of cycles prevents the appearance of this undesired product. After establishing an optimal cycle number, we performed 12 identical PCR1 reactions in order to amplify sufficient

amounts of genomic DNA and pooled them all prior to cleanup with a QIAquick PCR Purification Kit.

## XhoI digest

1 μg of the resulting DNA was digested with XhoI overnight at 37°C. Digest reactions consisted of 3.5 μL 10X FD buffer, 1 μL of 20,000 units/mL XhoI, 1 μg of DNA and dH2O to 35 μL. Heat inactivation of XhoI is not recommended, as the high temperatures result in reappearance of the spurious annealing products mentioned above, leading to a disruption of the XhoI overhang required for ligation. For the TRC1 and TRC1.5 libraries, there are two XhoI sites within the product of PCR1, resulting in fragments of 271, 43 and 183 bp. In order to purify the desired fragment, the entire digest was run on a 2% TAE agarose gel and purified the 271 bp fragment using a QIAquick Gel Extraction Kit. Once the band was excised, three volumes of buffer QG were added and the mixture heated at 30°C to dissolve the agarose. Lower melting temperatures are recommended so as not to denature the complementary double-stranded shRNA cassettes, which may not reanneal to their cognate strand. After the agarose was dissolved, one volume of isopropanol was added and protocol resumed following the manufacturer's instructions including the optional addition of NaOAc.

## Ligation of barcoded linkers

We prepared the barcoded linkers required for ligation by resuspending the lyophilized oligonucleotides in ST buffer (10 mM Tris pH 8.0, 50 mM NaCl) to 200 μM and combining 25 μL of each for a final concentration of 100 μM. The mixture was heated to 94°C for 10 min and gradually cooled to ensure proper annealing. Single-stranded oligonucleotides were removed from annealed oligonucleotides using Illustra MicroSpin G-25 columns. The sense (S1-S4) oligonucleotides are 5'-phosphorylated and the antisense oligonucleotides (AS1-AS4) each contain a single phosphorothioate bond at the 3' end to stabilize them and are designed to prevent the reformation of a functional XhoI site. The barcodes within these linkers are used for multiplexing and their length ensures they are compatible with the Illumina HiSeq 2000. Shorter barcode sequences may be compatible with other sequencing platforms. The selected barcoded linkers were added to ligation reactions with 100 ng of each purified 271 bp XhoI fragment, 3.5 μL 10X T4 DNA ligase buffer, 4 μL of 1 μM barcoded linker, 1 μL T4 DNA ligase and dH$_2$O to 35 μL. Ligations were performed overnight at 16°C. The entire ligation was run on a 2% TAE agarose gel and the resulting 312 bp band purified using the QIAquick Gel Extraction Kit in the same manner as previously described.

## PCR2

The final step in the preparation of the sequencing library is a second PCR with oligonucleotides that contain the Illumina adaptors required for bridge amplification and sequencing. In this PCR, the number of cycles is minimized in order to avoid PCR bias as well as errors that could affect sequencing. The reaction for PCR2 was as follows: 10 μL 5X Phusion HF buffer, 1 μL dNTPs (10 mM each), 2.5 μL Forward adapter primer (10 μM) 2.5 μL, Reverse adapter primer (10 μM), 1 μL Phusion DNA polymerase 10 ng barcoded DNA, and dH$_2$O to 50 μL. The cycling program consisted of 1 cycle of 98°C for 2 min, 2 cycles of 98°C for 30 s, 62°C for 30 s, 72°C for 30 s, 7 cycles of 98°C for 30 s, 72°C for 30 s and 1 cycle of 72°C for 3 min. The final 141 bp product was purified on a 2% TAE-agarose gel followed by QIAquick Gel Extraction as described above.

## Illumina sequencing

We assessed the purity of our sequencing library using the Bioanalyzer High Sensitivity DNA Kit (Agilent-5067-4626) and confirmed the presence of a single 141 bp peak, indicating one PCR product at the appropriate size. We utilized a multiplexing strategy consisting of four different barcodes with each nucleotide represented at each position of the barcode, allowing us to sequence four samples in each lane on a HiSeq 2000 Illumina instrument. To accomplish this, each sample was quantified and mixed together at a final concentration of 10 ng/μL and using Illumina-specific oligonucleotides and qPCR, we determined the cluster formation efficiency (i.e. effective concentration) of our library to be slightly greater than that of a known library. Accordingly, we loaded the flow cell at 5 pM and included a 10% ΦX-174 spike-in, which aids in quality control of cluster formation and sequencing on the Illumina platform. Cluster formation efficiency and the concentration of library to be loaded on

the flow cell needs to be determined empirically for each library preparation. These loading conditions yielded cluster densities between 733,000 clusters/mm$^2$ and 802,000 clusters/mm$^2$ and between 203 and 222 million reads per lane.

## shRNA screen analysis

shRNA data were analyzed in a similar fashion to RNA-seq data. Briefly, quality control was performed with FastQC, reads were trimmed to include only shRNA sequences using FASTQ trimmer, and filtered with the FASTQ Quality Filter. Reads were then aligned to a custom reference library of shRNA sequences using TopHat2. Three out of 36 samples were removed based on poor performance in unsupervised hierarchical clustering and/or principal component analysis, but each fibroblast cell line retained at least two biological replicates and nine of 12 retained all three replicates. Count tables were generated using HTSeq and differential expression determined by DESeq2.

## SOMAScan proteomics

Cell lysates from all 12 fibroblast cell lines were analyzed using SOMAscan v4.0 according to manufacturer's instructions and as previously reported (*Hathout et al., 2015*; *Mehan et al., 2014*). Data were analyzed using the QPROT statistical package (*Choi et al., 2015*).

## Isolation of RNA from LSK cells for RNA-seq

The whole bone marrow was harvested from the long bones of Dp16 mice (RRID:IMSR_JAX:013530) and matched littermate controls. Cells were first purified using hemolysis to remove RBCs and then stained and sorted for LSK cells (CD3-, Ter119-, Mac1-, Gr1-, B220-, Sca1+, cKit) using the Moflo XDP 70 FACS sorter. RNA was then isolated from these cells using the RNeasy Kit from Qiagen.

# Acknowledgements

This work was supported primarily by the Linda Crnic Institute for Down Syndrome and the Anna and John J Sie Foundation. We thank T Blumenthal and J Costello for stimulating discussion and critical reading of this manuscript. We also thank the individuals with Down syndrome that donated the biological samples that enabled these studies. We also thank the Functional Genomics, Genomics, and Flow Cytometry Shared Resources at the University of Colorado Cancer Center.

# Additional information

## Competing interests

JME: Reviewing editor, *eLife*. The other authors declare that no competing interests exist.

## Funding

| Funder | Grant reference number | Author |
| --- | --- | --- |
| University of Colorado | Linda Crnic Institute for Down Syndrome | Joaquín M Espinosa |
| Howard Hughes Medical Institute | | Joaquín M Espinosa |
| National Institutes of Health | R01CA117907 | Joaquín M Espinosa |
| National Science Foundation | MCB-1243522 | Joaquín M Espinosa |
| Anna and John J. Sie Foundation | | Joaquín M Espinosa |
| National Institutes of Health | P30CA046934-27 | Joaquín M Espinosa |

The funders had no role in study design, data collection and interpretation, or the decision to submit the work for publication.

## Author contributions

KDS, HCL, AAH, LPJ, KPS, LAL, Conception and design, Acquisition of data, Analysis and interpretation of data, Drafting or revising the article; AP, MDG, JD, JME, Conception and design, Analysis

and interpretation of data, Drafting or revising the article; JMC, Conception and design, Acquisition of data, Analysis and interpretation of data; EBG, Conception and design, Acquisition of data

### Author ORCIDs

Joaquín M Espinosa, http://orcid.org/0000-0001-9048-1941

### Ethics

Human subjects: All individuals in this study were consented on Colorado Multiple Institutional Review Board (COMIRB)-approved protocols (Protocol Numbers: 11-1790 or 15-1774) and samples collected and processed as describe in the Materials and Methods.

Animal experimentation: This study was performed in strict accordance with the recommendations in the Guide for the Care and Use of Laboratory Animals of the National Institutes of Health. All of the animals were handled according to approved institutional animal care and use committee (IACUC) protocols of the University of Colorado. The protocol was approved by the University of Colorado IACUC (Protocol Number: B-41413(04)1E).

# Additional files

### Supplementary files

• Supplementary file 1. (A) Fibroblast, (B) lymphoblastoid, (C) Dp16, (D) monocyte, (E) T cell and (F) meta RNA-seq. DESeq2 analysis of T21 versus D21 fibroblasts. Columns include: (A) Chromosome, (B) Gene start coordinate, (C) Gene end coordinate, (D) Gene strand, (E) Gene name, (F) basemean (average read count across all samples), (G) basemeanD21 (average read count across all D21 samples), (H) basemeanT21 (average read count across all T21 samples), (I) foldChange (basemeanT21/basemeanD21), (J) log2FoldChange, (K) foldChange_adj (DESeq2 adjusted fold change), (L) log2FoldChange_adj, (M) pval (p-value), (N) padj (Benjamini-Hochberg adjusted p-value).

• Supplementary file 2. Fibroblast kinome shRNA screen analysis. DESeq2 analysis of kinome shRNA screens in T21 versus D21 fibroblasts. Columns include: (A) TRC number (B) shRNA targeting location (C) Chromosome, (D) Genomic coordinates, (E) Gene strand, (F) Gene name, (G) RefSeq ID (H) basemean (average read count across all samples), (I) basemeanD21 (average read count across all D21 samples), (J) basemeanT21 (average read count across all T21 samples), (K) foldChange (basemeanT21/basemeanD21), (L) log2FoldChange, (M) foldChange_adj (DESeq2 adjusted fold change), (N) log2FoldChange_adj, (O) pval (p-value), (P) padj (Benjamini-Hochberg adjusted p-value).

• Supplementary file 3. Fibroblast SOMAscan analysis. QPROT analysis of T21 versus D21 fibroblasts. Columns include: (A) Chromosome, (B) Gene start coordinate, (C) Gene end coordinate, (D) Gene strand, (E) Gene name, (F) RFUmean (average RFU across all samples), (G) RFUmeanD21 (average RFU across all D21 samples), (H) RFUmeanT21 (average RFU across all T21 samples), (I) foldChange (RFUmeanT21/RFUmeanD21), (J) log2FoldChange, (K) Zstatistic (Z-score from QPROT), (L) FDRup (FDR of upregulated proteins), (M) FDRdown (FDR of downregulated proteins).

### Major datasets

The following datasets were generated:

| Author(s) | Year | Dataset title | Dataset URL | Database, license, and accessibility information |
|---|---|---|---|---|
| Sullivan KD, Pandey A, Jackson LP, Espinosa JM | 2016 | RNAseq from disomic and trisomic Fibroblasts and Lymphoblasts | http://www.ncbi.nlm.nih.gov/geo/query/acc.cgi?acc=GSE79842 | Publicly available at NCBI Gene Expression Omnibus (accession no: GSE79842) |

| | | | | |
|---|---|---|---|---|
| Sullivan KD, Pandey A, Jackson LP, Espinosa JM | 2016 | shRNA cassette sequencing from disomic and trisomic fibroblasts cultured in the presence of shRNA kinome library for 14 days | http://www.ncbi.nlm.nih.gov/geo/query/acc.cgi?acc=GSE79840 | Publicly available at NCBI Gene Expression Omnibus (accession no: GSE79840) |
| Sullivan KD, Pandey A, Smith KP, Espinosa JM | 2016 | RNAseq from disomic and trisomic T cells and monocytes | http://www.ncbi.nlm.nih.gov/geo/query/acc.cgi?acc=GSE84531 | Publicly available at NCBI Gene Expression Omnibus (accession no: GSE84531) |
| Liggett LA, Sullivan KD, Pandey A | 2016 | RNAseq from Dp16 and control mice | http://www.ncbi.nlm.nih.gov/geo/query/acc.cgi?acc=GSE84526 | Publicly available at NCBI Gene Expression Omnibus (accession no: GSE84526) |

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
