## [Decision Letter]

Thank you for submitting your article "Trisomy 21 consistently activates the interferon response leading to decreased cellular fitness" for consideration by *eLife*. Your article has been favorably evaluated by Charles Sawyers (Senior editor) and three reviewers, one of whom is a member of our Board of Reviewing Editors. The reviewers have opted to remain anonymous.

Summary:

This paper addresses the effects of age, gender and ethnicity on expression of chromosome 21 genes in Down syndrome and leverages a clever screen to identify effectors of growth suppression in DS. The study raises the interesting possibility of using ruxolitinib to treat DS phenotypes. The paper also convincingly supports the conclusions of Do et al. (2015) that changes in gene expression in cells with trisomy 21 occur throughout the genome and are not confined to specific domains. The paper was viewed enthusiastically by a member of the BRE and two additional referees. The BRE commented, 'These findings have important implications for understanding the basic nature of, and individual variation in, the diverse phenotypes of people with Down's syndrome'. One referee noted, 'Overall the paper is an interesting and significant study that provides important new insights into the differences in gene expression between D21 and T21 cells'. The second referee commented, 'In principle, the findings are tremendously exciting and present a novel view of the pathology associated with trisomy 21.'

Essential revisions:

The reviewers agreed that the following important additional points need to be addressed.

1) Given the potential enormous impact of the findings, it is incumbent upon the authors to verify at least some of their findings on primary cells at the protein level. There are adequate reagents to establish expression of IFN receptors, pSTAT1 and various IFN-regulated genes in primary cells (e.g. lymphocytes, monocytes etc.) in a subset of subjects with trisomy 21 versus controls. Minimally, this should be done with the fibroblast and lymphoblastoid cell lines including appropriate controls, with and without addition of exogenous IFN (Figure 4 for instance lacks normal controls). If possible, the relevance of these findings would be greatly advanced by any evidence for enhanced activation of interferon signaling in primary blood cells or serum/plasma of individuals with Down syndrome compared to control individuals. Alternatively, the authors might consider using the Ts65Dn murine model of trisomy 21. It should be simple to analyze pSTAT1 and gene expression in leukocytes from these mice compared to controls. Clearer documentation of the dysregulation of IFNR expression and consequences in primary cells would help the authors' argument and strengthen the very provocative and exciting conclusions.

2) The nature of the screen is a bit confusing as written. It is difficult to figure out what the actual assay was for the differential requirements of the kinases. This could be better described. Also, a major concern is that there are no data regarding the differences in growth between D21 and T21 cells. Data from proliferation and apoptosis assays should be included. How different are the growth rates? Finally, the text suggests that ruxolitinib promoted the growth of T21 cells, but the data in Figure 4 show viability. Are there data to support changes in proliferation or is it just survival that is affected by JAK inhibition?

3) The increase in IFN gene expression and the sensitivity to ruxolitinib suggests that there is a significant change in JAK/STAT signaling in the T21 cells. Western blots comparing the levels of STAT activation in the T21 versus D21 cells should be included.

4) Ruxolitinib was approved for the treatment of myelofibrosis and for some cases of polycythemia vera. So far, it has not shown significant activity in acute leukemia. It should be noted that the dose limiting toxicities of ruxolitinib are anemia and thrombocytopenia. There may be a therapeutic index in which the drug will have a therapeutic benefit in DS individuals and not be toxic, but there may not be one. This issue should be mentioned in the Discussion to prevent lay people from becoming aware of these findings and thinking that ruxolitinib would ameliorate DS phenotypes without side effects and without further studies.

5) The connection between mTOR and the differential growth of D21 and T21 isn't obvious. This could be discussed in greater depth. Note, however, that the Liu paper on neonatal megakaryocytes doesn't address differences in a DS background. They refer to TMD because of its neonatal presentation, but don't directly study in cells with trisomy 21.

---

## [Author Response]

Essential revisions:

*The reviewers agreed that the following important additional points need to be addressed.*

*1) Given the potential enormous impact of the findings, it is incumbent upon the authors to verify at least some of their findings on primary cells at the protein level. There are adequate reagents to establish expression of IFN receptors, pSTAT1 and various IFN-regulated genes in primary cells (e.g. lymphocytes, monocytes etc.) in a subset of subjects with trisomy 21 versus controls. Minimally, this should be done with the fibroblast and lymphoblastoid cell lines including appropriate controls, with and without addition of exogenous IFN (Figure 4 for instance lacks normal controls). If possible, the relevance of these findings would be greatly advanced by any evidence for enhanced activation of interferon signaling in primary blood cells or serum/plasma of individuals with Down syndrome compared to control individuals. Alternatively, the authors might consider using the Ts65Dn murine model of trisomy 21. It should be simple to analyze pSTAT1 and gene expression in leukocytes from these mice compared to controls. Clearer documentation of the dysregulation of IFNR expression and consequences in primary cells would help the authors' argument and strengthen the very provocative and exciting conclusions.*

We thank the reviewers for these comments and are happy to report that the revised manuscript describes several lines of evidence demonstrating that dysregulation of IFN signaling is not restricted to the transcriptome of cell lines grown in vitro, but that is also evident in primary cells –both human and murine– and manifested at the protein level as well.

First, during the review period we were able to collect 17 blood samples, 10 of them from individuals with trisomy 21. Similar to our initial work using cell lines, we made sure to obtain samples from individuals of both genders, different ages, and different genetic backgrounds. We then processed the blood samples to purify monocytes, T cells, and B cells. We discarded the possibility of simply looking at peripheral blood mononucleated cells (PBMCs) because there is ample evidence in the literature that individuals with trisomy 21 display changes in the numbers of various blood cell types, which could confound the interpretation of any gene expression changes observed as driven by cell type abundance rather than actual gene regulation. Hence, we opted to analyze more defined cell populations within the hematopoietic system. Due to the limited volume of blood available, we maximized our chances to detect consistent gene expression changes caused by trisomy 21 by performing a combination of assays to analyze both RNA and protein expression. Given the high sensitivity of transcriptome analyses via next-generation sequencing, we decided to analyze monocytes and T cells with RNA-seq. B cell populations were used for Western blot and flow cytometry assays. The RNA-seq experiments were very successful, producing high quality data, and demonstrated unequivocally that IFN signaling is constitutively activated in both monocytes and T cells. As previously observed in fibroblasts and lymphoblasts, the gene expression signature associated with trisomy 21 has clear signs of activation of the interferon transcriptional response. Very interestingly, in addition to the expected upregulation of the IFN receptors and many IFN-stimulated genes (ISGs), transcriptome analysis of blood cell types revealed another hallmark of IFN hyperactivation that was not obvious in the analysis of in vitro cell cultures: strong downregulation of the translation machinery. It is well established that a key consequence of the IFN response is to prevent synthesis of viral proteins (Walsh et al. 2013; Gupta et al. 2012; Ivashkiv & Donlin 2014). However, the mechanisms by which IFN signaling affects translation are not well understood. RNA-seq analysis of monocytes and T cells revealed downregulation of dozens of translation initiation and elongation factors, as well as ribosomal proteins in both the small and large ribosome subunits. The importance of this result cannot be overstated, as repression of protein synthesis in tissues from individuals with Down syndrome could contribute to many of the ill effects of trisomy 21. We decided that this important result should occupy a central place in the manuscript, and hence created a new main figure with two supplements to clearly explain these results (Figure 7, Figure 7—figure supplement 1 and Figure 7—figure supplement 2 in the revised manuscript). Unfortunately, our efforts using western blot for analysis of B cells were not successful, due mostly to the limited amount of protein obtained, but we did manage to measure surface expression of four IFN receptors -three encoded on chr21 (IFNAR1, IFNGR2, IL10RB) and one not encoded on chr21 (IFNGR1)- using flow cytometry for all 17 samples. Expectedly, these experiments revealed a modest but reproducible increase in the expression of the three IFNRs encoded on chr21 (Figure 6—figure supplement 2). As described below, our western blot efforts produced very clear results when employing cell lines, which could be grown in large amounts.

Second, in order to verify that increased IFN signaling does not occur only at the transcriptome level, we embarked on a massive effort to analyze changes at the protein level. As suggested by Reviewers, we employed conditions of both basal and stimulated IFN signaling. These efforts resulted in the inclusion of 59 western blot panels analyzing 90 different protein extracts in the revised manuscript. Analysis of basal protein expression in 12 fibroblast cell lines and 6 lymphoblastoid cell lines demonstrated that the IFN receptors are indeed upregulated at the protein level, along with increased levels of phospho-STAT1 and several ISGs, with the expected inter-individual variation observed during the transcriptome analysis (see Figure 2 for basal expression in fibroblasts and Figure 5 for basal expression in lymphoblastoids). Furthermore, we analyzed protein lysates from the 12 fibroblast cell lines using SOMAscan technology, which employs DNA aptamers to monitor protein abundance (Hathout et al. 2015). This effort confirmed induction of many interferon-related genes at the protein level (see Figure 2). Additionally, we performed a battery of dose-response experiments with three IFN ligands (IFN-α, IFN-β and IFN-γ) on three different pairs of euploid/trisomy 21 fibroblast cell lines. This effort clearly demonstrated that the induced expression of the ISGs tested (MX1, IDO1 and ISG15) is also greater in trisomy 21 cells at the protein level, with signs of inter-individual variation for specific genes, as predicted by the RNA-seq experiments. This comprehensive analysis of protein expression is shown in a new Figure 3.

Third, following Reviewers’ suggestions, we also analyzed primary cells from a mouse model of Down syndrome. We chose to use the Dp16 strain, which carries triplication of only genes found on human chr21. The Ts65DN model proposed by the reviewers has triplication of many genes not on chr21. Of note, the chromosome fragment triplicated in the Dp16 strain contains the four IFN receptors found on human chr21 (Li et al. 2007). RNA-seq analysis of LSK (Lineage negative, Sca1 positive, c-Kit positive) cells confirmed that the IFN receptors are overexpressed in Dp16 mice relative to disomic littermates, as well as several canonical ISGs. These results are now included in Figure 5—figure supplement 3.

*2) The nature of the screen is a bit confusing as written. It is difficult to figure out what the actual assay was for the differential requirements of the kinases. This could be better described. Also, a major concern is that there are no data regarding the differences in growth between D21 and T21 cells. Data from proliferation and apoptosis assays should be included. How different are the growth rates? Finally, the text suggests that ruxolitinib promoted the growth of T21 cells, but the data in Figure 4 show viability. Are there data to support changes in proliferation or is it just survival that is affected by JAK inhibition?*

We thank these comments from the Reviewers and acknowledge that the shRNA screen results could have been better explained. The revised manuscript addresses this comment in several ways. First, we have expanded the description of the screen and the interpretation of the results in the Results section. Second, we performed additional experiments to investigate in more detail the effects of JAK inhibition. Third, we displayed the results in a more accessible fashion. The revised Figure (now Figure 4 and Figure 4—figure supplement 1), displays the results of analyzing two pairs of euploid/trisomic fibroblasts during a dose-response of Ruxolitinib. Expectedly, pharmacological JAK inhibition leads to decreased phospho-STAT levels (which is higher in the T21 cell lines) and decreased expression of the IFN-stimulated gene MX1. Strikingly, in the absence of JAK inhibition, there is a significant decrease in the number of trisomy 21 cells in the absence of any treatment. Addition of increasing doses of ruxolitinib enhances the number of viable trisomic cells while decreasing the number of their viable disomic counterparts. Consequently, the relative number of trisomic versus disomic cells increases with the dose of Ruxolitinib. This would explain why, over the course of the two weeks of shRNA screening, shRNAs targeting JAK1 (and TYK2) would show a relative enrichment in trisomy 21 cells.

*3) The increase in IFN gene expression and the sensitivity to ruxolitinib suggests that there is a significant change in JAK/STAT signaling in the T21 cells. Western blots comparing the levels of STAT activation in the T21 versus D21 cells should be included.*

The requested phospho-STAT1 Western blots are now included at various places throughout the manuscript. Increased levels of basal phospho-STAT1 are shown in Figure 2, Figure 4, Figure 4—figure supplement 4D, and Figure 5. Increased levels of induced phospho-STAT1 in trisomy 21 cells are evident for some fibroblast pairs in Figure 3. Of note, all three IFN ligands consistently induced STAT1 phosphorylation (pSTAT1) both in D21 and T21 cells, but the levels of pSTAT1 did not correlate precisely with the expression levels of the various ISGs. For example, the obviously different levels of ISG15 upon treatment with the three ligands do not correlate with dissimilar levels of pSTAT1 (see cell line pair 2 in Figure 3). This suggests that STAT1 phosphorylation is not a robust predictor of ISG expression, which is ultimately defined by the orchestrated action of multiple IFN-activated transcription factors. Finally, we have now included western blots showing dose-dependent repression of pSTAT1 signaling and MX1 expression in response to ruxolitinib treatment for two T21-D21 cell line pairs. We have described these findings in detail in the Results section.

*4) Ruxolitinib was approved for the treatment of myelofibrosis and for some cases of polycythemia vera. So far, it has not shown significant activity in acute leukemia. It should be noted that the dose limiting toxicities of ruxolitinib are anemia and thrombocytopenia. There may be a therapeutic index in which the drug will have a therapeutic benefit in DS individuals and not be toxic, but there may not be one. This issue should be mentioned in the Discussion to prevent lay people from becoming aware of these findings and thinking that ruxolitinib would ameliorate DS phenotypes without side effects and without further studies.*

We fully agree with the need for caution and much further investigation before ruxolitinib or any other Interferon antagonist could be used in the clinical setting to ameliorate the ill effects of trisomy 21. The Discussion thoroughly addresses the comment from the Reviewers.

*5) The connection between mTOR and the differential growth of D21 and T21 isn't obvious. This could be discussed in greater depth. Note, however, that the Liu paper on neonatal megakaryocytes doesn't address differences in a DS background. They refer to TMD because of its neonatal presentation, but don't directly study in cells with trisomy 21.*

We thank this comment form the Reviewers, which we address in the revised manuscript in a number of ways. We now include the data for shRNAs for mTOR in Figure 4 and discuss the reported hyperactivation of mTOR in both human and mouse systems, citing additional references.